# IPFormer: Visual 3D Panoptic Scene Completion with Context-Adaptive Instance Proposals

**Markus Gross**[1,2,3,*]     **Aya Fahmy**[1]     **Danit Niwattananan**[2]
**Dominik Muhle**[2,3]     **Rui Song**[4]     **Daniel Cremers**[2,3]     **Henri Meeß**[1]

[1]Fraunhofer Institute IVI
[2]Technical University of Munich
[3]Munich Center for Machine Learning
[4]University of California, Los Angeles

## Abstract

Semantic Scene Completion (SSC) has emerged as a pivotal approach for jointly learning scene geometry and semantics, enabling downstream applications such as navigation in mobile robotics. The recent generalization to Panoptic Scene Completion (PSC) advances the SSC domain by integrating instance-level information, thereby enhancing object-level sensitivity in scene understanding. While PSC was introduced using LiDAR modality, methods based on camera images remain largely unexplored. Moreover, recent Transformer-based approaches utilize a fixed set of learned queries to reconstruct objects within the scene volume. Although these queries are typically updated with image context during training, they remain static at test time, limiting their ability to dynamically adapt specifically to the observed scene. To overcome these limitations, we propose IPFormer, the first method that leverages context-adaptive instance proposals at train and test time to address vision-based 3D Panoptic Scene Completion. Specifically, IPFormer adaptively initializes these queries as panoptic instance proposals derived from image context and further refines them through attention-based encoding and decoding to reason about semantic instance-voxel relationships. Extensive experimental results show that our approach achieves state-of-the-art in-domain performance, exhibits superior zero-shot generalization on out-of-domain data, and achieves a runtime reduction exceeding $14\times$. These results highlight our introduction of context-adaptive instance proposals as a pioneering effort in addressing vision-based 3D Panoptic Scene Completion. Code available at https://github.com/markus-42/ipformer.

## 1   Introduction

Panoptic Scene Completion provides a holistic scene understanding that can serve applications like autonomous driving and robotics by reconstructing volumetric geometry from sparse sensor data and assigning meaning to objects in the scene. Such holistic scene understanding requires both accurate geometry and semantic information about a scene. Historically, these tasks have been treated separately and evolved independently due to the distinct nature of geometric reconstruction and semantic interpretation, each requiring specialized algorithms and data representations. Despite the inherent interconnection between geometry and semantics, their separation hindered unified scene understanding. Classical structure-from-motion evolved into simultaneous localization and mapping (SLAM) systems for accurate reconstructions [38, 12, 25], while deep learning facilitated monocular depth estimation (MDE) methods [14, 15, 53], which are restricted to visible surfaces. In contrast,

---

[*]Corresponding author: markus.gross@tum.de

39th Conference on Neural Information Processing Systems (NeurIPS 2025).

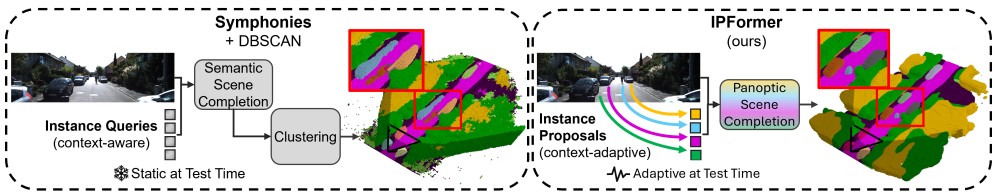

Figure 1: Comparison of query initialization for Panoptic Scene Completion (PSC). Existing methods, *e.g.* Symphonies [21], randomly initialize instance queries and incorporate context-awareness during training. However, these queries retain their static nature at test time, as they are shared across all inputs. Our method IPFormer initializes them as *instance proposals*, which preserve context-adaptivity at test time, effectively aggregating directed features for improved PSC performance. Due to the the novelty of vision-based PSC and the absence of established baselines, we apply DBSCAN [13] clustering to Symphonies' SSC output to retrieve its individual instances.

Scene Completion refers to the process of inferring the complete 3D structure of a scene, including both visible and occluded regions, from partial observations [52, 16].

Apart from geometry, semantic image segmentation divides an image into semantically meaningful regions and labels them [58, 7]. Panoptic segmentation [23, 22, 41] extends this approach by segmenting instances [17] for *things* (countable objects like person or car) while maintaining semantic segmentation for *stuff* (amorphous regions like road or vegetation). This unified approach is vital for comprehensive scene understanding, as it distinguishes individual object instances while preserving semantic context for background elements, enabling applications like autonomous navigation. The recent trend in scene understanding has been to unify these tasks into Semantic Scene Completion (SSC) [19] and Panoptic Scene Completion (PSC) [3]. SSC integrates geometric completion with semantic labeling, predicting both the 3D structure and semantic categories, represented as a 3D grid composed of voxels. PSC builds on this by adding instance-level segmentation for each voxel, thus distinguishing between individual objects. While PSC was introduced using LiDAR modality, methods based on camera images remain largely unexplored. Moreover, recent Transformer-based approaches utilize a fixed set of learned queries to reconstruct objects within the scene volume. Although these queries are typically updated with image context during training, they remain static at test time, limiting their ability to dynamically adapt specifically to the observed scene.

To address these limitations, we introduce IPFormer, a 3D Vision Transformer designed to address Panoptic Scene Completion. IPFormer lifts contextual 2D image features to 3D and samples them based on visibility to initialize instance proposals, each implicitly representing semantic instances within the camera view. Based on this context-adaptive initialization, we establish a robust reconstruction signal that enhances subsequent encoding and decoding stages to reconstruct a complete panoptic scene. Sampling instance proposals solely from visible surfaces is grounded in the principle that all potentially identifiable objects within the observed scene must exhibit a visual cue in the camera image to facilitate completion. This approach of directed feature aggregation at train and test time significantly improves instance identification, semantic classification, and geometric completion.

Our **contributions** can be summarized as follows:

- We present IPFormer, the first method that leverages context-adaptive instance proposals at train and test time to address vision-based 3D Panoptic Scene Completion.

- We introduce a visibility-based sampling strategy, which initializes instance proposals that dynamically adapt to scene characteristics, improving PQ-All by $3.62\%$ and Thing-metrics on average by $18.65\%$, compared to non-adaptive initialization.

- Our experimental results demonstrate that IPFormer achieves state-of-the-art performance on in-domain data, exhibits superior zero-shot generalization on out-of-domain data, and offers a runtime reduction of over $14\times$, from $4.51$ seconds to $0.33$ seconds.

- Comprehensive ablation studies reveal that employing a dual-head architecture combined with a two-stage training strategy, in which SSC and PSC tasks are trained independently and sequentially, significantly improves performance and effectively balances metrics.

## 2 Related work

**Semantic Scene Completion.** The task of Semantic Scene Completion has seen a large amount of interest after it was first introduced in [44]. The initial work focused on solving the task for indoor scenes from camera data [60, 61, 35, 29, 28, 27, 8, 2] and used point clouds in outdoor scenarios with LiDAR-based methods [45, 9, 57, 30, 43]. In the following, we will discuss the most relevant contributions for Semantic Scene Completion in the context of autonomous driving. For a more thorough overview, we refer the reader to the surveys of [56] and [62].

Recent progress in perception for autonomous driving has increasingly adopted Vision Transformer architectures [11], notably DETR [5]. DETR introduced learnable object queries, which are trainable embeddings that allow the model to directly predict object sets in parallel, eliminating hand-crafted components like anchor boxes. In SSC, learnable queries can be adapted to predict semantic labels for 3D scene elements (*e.g.*, voxels), streamlining the process and enhancing efficiency. Architectures like OccFormer [63] and VoxFormer [32] exemplify this trend, with OccFormer using multiscale voxel representations with query features to decode 3D semantic occupancy, and VoxFormer employing depth-based queries fused with image features, refined via deformable attention. Similarly, Symphonies [21] refines instance queries iteratively using cross-attention between image features and 3D representations, while TPVFormer [20] leverages a tri-perspective view (TPV) to query scene features efficiently. HASSC [50] and CGFormer [59] further refine this general approach by focusing on challenging voxels and contextual queries, respectively. Alternative approaches, such as implicit scenes via self-supervised learning in S4C, [19] and CoHFF [48], which leverages both voxel and TPV representations to fuse features from multiple vehicles, have also emerged, though they do not utilize learnable queries.

It is worth noting that existing literature presents ambiguity regarding context-awareness of queries for SSC. Although both Symphonies [21] and CGFormer [59] incorporate contextual cues during training, their approaches diverge at inference. In particular, Symphonies' instance queries remain static and are shared across all input images, whereas CGFormer's voxel queries preserve context-awareness by dynamically adapting to each input. To clarify this distinction, we differentiate between **instance queries** and **instance proposals** as context-aware and context-adaptive, respectively. Our introduced *instance proposals* effectively adapt scene characteristics to guide feature aggregation during both train and test time.

**Panoptic Scene Completion.** While panoptic LiDAR segmentation, which predicts panoptics for point clouds directly, has been widely studied [6, 55, 54], voxel-based PSC approaches have only recently been introduced by PaSCo [3]. This method employs a hybrid convolutional neural network (CNN) and Transformer architecture with static instance queries to address LiDAR-based PSC, enhanced by an uncertainty-aware framework. In parallel, camera-based PSC from multi-frame and multi-view (surround-view) images is addressed by PanoOcc [51], which utilizes static voxel queries to aggregate spatiotemporal information, while SparseOcc [34] introduces a sparse voxel decoder and sparse static instance queries to predict semantic and instance occupancy from up to 96 temporal frames. Concurrently, PanoSSC [47] introduces a camera-based PSC method using forward-facing, non-temporal imagery. Its dual-head TPV-based architecture separates semantic occupancy and instance segmentation tasks to infer a 3D panoptic voxel grid. Likewise, our method targets PSC using forward-facing, non-temporal imagery. However, it overcomes a key limitation of prior approaches by avoiding reliance on static queries and instead introduces context-adaptive instance proposals that dynamically adapt to the scene at both train and test time. Notably, PanoSSC [47] is trained on a non-public post-processed version of the SemanticKITTI dataset [1], which, in addition to the absence of released source code, challenges direct comparisons.

## 3 Methodology

### 3.1 Overview

Given an input image $\mathbf{X} \in \mathbb{R}^{U \times V \times 3}$ (Fig. 2), the backbone predicts image features $\mathbf{F} \in \mathbb{R}^{H \times W \times F}$, and the depth net predicts a depth map $\mathbf{D} \in \mathbb{R}^{U \times V \times 1}$, following VoxFormer [32]. Further refining the image features and the depth map allows for generating context-aware 3D representations, denoted as 3D context features $\mathbf{F}_{3D}$ and initial voxels $\mathbf{V}$. Subsequently, visible and invisible voxels, sampled from $\mathbf{V}$ using the depth map, attend to the 3D image features $\mathbf{F}_{3D}$ via a series of 3D deformable cross

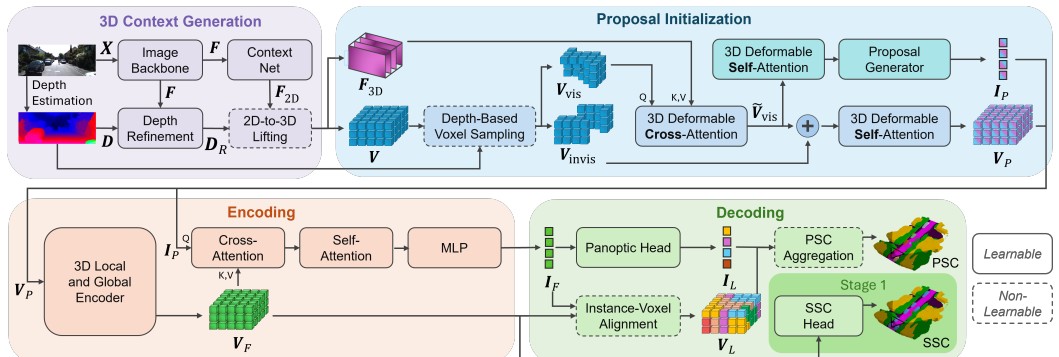

Figure 2: Detailed architecture of IPFormer. Our method refines image features and a depth map to produce 3D context features, which are sampled based on visibility to generate context-adaptive instance and voxel proposals. In a two-stage training strategy, voxel proposals first handle Semantic Scene Completion, guiding the latent space toward detailed geometry and semantics. The second stage attends instance proposals over the pretrained voxel features to register individual instances. This dual-head design aligns semantics, instances and voxels, enabling robust Panoptic Scene Completion.

and self-attention mechanisms. This process initializes instance proposals $\mathbf{I}_P$ and voxel proposals $\mathbf{V}_P$. In this setup, every instance proposal can eventually correlate with a single instance within the observed scene.

The voxel proposals $\mathbf{V}_P$ are encoded by 3D local and global encoding, in line with CGFormer [59], resulting in voxel features $\mathbf{V}_F$. We follow a two-stage training strategy in which the first stage uses these voxel features $\mathbf{V}_F$ to address SSC exclusively. This approach effectively guides the latent space towards semantics and geometry. The second training stage targets overall PSC, thereby detecting individual instances. This is achieved by attending the instance proposals $\mathbf{I}_P$ over the pre-trained voxel features from the first stage, resulting in instance features $\mathbf{I}_F$. These features are used to (i) predict semantics, and (ii) align instances and voxels to aggregate a complete panoptic scene.

## 3.2 3D Context Generation

Following CGFormer [59], we refine the depth map $\mathbf{D}$ and the feature map $\mathbf{F} \in \mathbb{R}^{H \times W \times F}$ into per-pixel, frustum-shaped depth probabilities $\mathbf{D}_R \in \mathbb{R}^{H \times W \times D}$, effectively resulting in $|D|$ depth bins per pixel. This is achieved by applying $\Phi_D$ as a mixture of convolution and neighborhood cross-attention layers [18], formulated as $\mathbf{D}_R = \Phi_D(\mathbf{D}, \mathbf{F}_{2D})$. Moreover, the generic image features $\mathbf{F}$ are projected into contextual features $\mathbf{F}_{2D} \in \mathbb{R}^{H \times W \times C}$ through the context net $\Phi_C$, denoted as $\mathbf{F}_{2D} = \Phi_C(\mathbf{F})$.

The next step is to lift context features $\mathbf{F}_{2D}$ to 3D, following the design of [40]. To elaborate, we lift them to (i) 3D context features $\mathbf{F}_{3D}$ and (ii) 3D initial voxels $\mathbf{V}$. To achieve this, we distribute $\mathbf{F}_{2D}$ along rays defined by the camera intrinsics and weighted by the depth probability distribution $\mathbf{D}_R$. This is formulated as

$$\mathbf{F}_{3D}(u, v, d, c) = \mathbf{F}_{2D}(u, v, c) \cdot \mathbf{D}_R(u, v, d) , \tag{1}$$

where $(u, v)$ are pixel coordinates, $d$ indexes the depth bin, and $c$ is the feature channel dimension. This approach distributes feature vectors across depth bins according to their corresponding probability distributions and thus enables effective lifting of 2D features into a probabilistic 3D volume, while preserving both contextual and spatial information. Note that Eq. (1) does not include learnable parameters.

To additionally create discretized initial voxels $\mathbf{V} \in \mathbb{R}^{X \times Y \times Z \times C}$, we voxelize $\mathbf{F}_{3D}$ to a regular grid. Let $S = \{(u, v, d) | \mathcal{Q}((u, v, d, c)) = (x, y, z)\}$, then

$$\mathbf{V}(x, y, z) = \sum_{(u, v, d) \in S} \mathbf{F}_{3D}(u, v, d) , \tag{2}$$

where $\mathcal{Q}$ quantizes continuous coordinates to discrete voxel indices, and $C$ represents the feature dimension. The summation aggregates all features that map to the same voxel, effectively accumulating

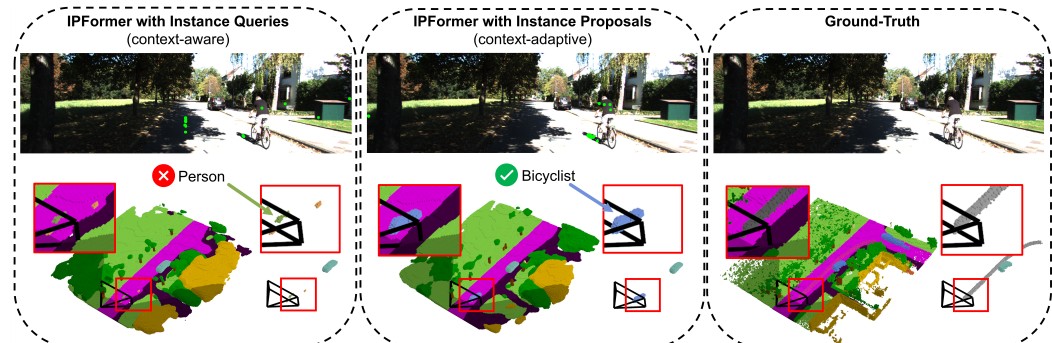

Figure 3: Instance-specific saliency. Through gradient-based attribution, we derive saliency maps that highlight image regions in green, where an individual instance mainly retrieves context from. Our introduced instance proposals effectively adapt to scene characteristics by guiding feature aggregation, substantially improving identification, classification, and completion. In contrast, instance queries sample context in an undirected manner, causing misclassification and geometric ambiguity.

evidence for stronger feature representations. Building upon the concept of [59], we add learnable embeddings to $\mathbf{V}$ that enhance its representational capacity.

### 3.3 Proposal Initialization

**Voxel Visibility.** The initial voxels $\mathbf{V}$ are categorized into visible voxels $\mathbf{V}_{\text{vis}}$ and invisible voxels $\mathbf{V}_{\text{invis}}$, following established SSC works such as [32]. This is achieved by first applying inverse projection $\Pi^{-1}$ on the depth map $\mathbf{D}$ with camera intrinsics $\mathbf{K} \in \mathbb{R}^{4 \times 4}$. This process generates a pseudo point cloud by unprojecting a pixel $(u, v)$ to a corresponding 3D point $(x, y, z)$. Furthermore, let $\mathbf{M} \in \mathbb{R}^{X \times Y \times Z}$ be a binary mask derived from this point cloud that filters the voxel features based on occupancy. Thus, the visible voxel features are defined as $\mathbf{V}_{\text{vis}} = \mathbf{V} \odot \mathbf{M}$, where $\mathbf{M}(x, y, z) = 1$ if $(x, y, z)$ corresponds to an unprojected point, and 0 otherwise. The operator $\odot$ denotes the Schur product. Consequently, the invisible voxel features are given by $\mathbf{V}_{\text{invis}} = \mathbf{V} \odot (\mathbf{1} - \mathbf{M})$, with all-ones matrix $\mathbf{1} \in \mathbb{R}^{X \times Y \times Z}$. Note that this sampling does not include learnable parameters.

To further adapt $\mathbf{V}_{\text{vis}}$ with image context, we apply a tailored series of 3D deformable attention mechanisms, inspired by [21, 59]. Specifically, we apply 3D deformable cross-attention (DCA) to $\mathbf{V}_{\text{vis}}$ (as queries) and the 3D context features $\mathbf{F}_{3D}$ (as keys and values), resulting in updated visible voxel features $\tilde{\mathbf{V}}_{\text{vis}}^{\mathbf{x}}$. More precisely, an updated voxel query $\tilde{\mathbf{V}}_{\text{vis}}^{\mathbf{x}}$ at 3D location $\mathbf{x}$ computes

$$\tilde{\mathbf{V}}_{\text{vis}}^{\mathbf{x}} = \text{DCA}(\mathbf{V}_{\text{vis}}, \mathbf{F}_{3D}, \mathbf{x}) = \sum_{n=1}^{N} A_n W \psi(\mathbf{F}_{3D}, \Pi(\mathbf{x}) + \Delta\mathbf{x}) , \tag{3}$$

where $\Pi(\mathbf{x})$ obtains the reference points, $\Delta\mathbf{x} \in \mathbb{R}^3$ denotes the estimated displacement from the reference point $\mathbf{x}$, and $\psi(\cdot)$ refers to trilinear interpolation applied to sample from the 3D context features $\mathbf{F}_{3D}$. The index $n$ loops through the sampled points out of a total of $N$ points, $A_n \in [0, 1]$ represents the attention weights, and $W$ signifies the transformation weight. We present only the formulation of single-head attention and we utilize multiple layers of deformable cross-attention.

**Instance Proposals.** The visible voxels $\tilde{\mathbf{V}}_{\text{vis}}$ are further processed to initialize context-adapted instance proposals $\mathbf{I}_P$. Recall from Sec. 1 that instance proposals are initialized at this stage of the architecture, because in principle, all potentially detectable instances within the observed scene must exhibit a visual cue in the camera image to facilitate their completion, thereby defining panoptic scene *completion*. Following this line of thought, we first apply 3D deformable self-attention (DSA) on $\tilde{\mathbf{V}}_{\text{vis}}$ to foster global context-aware attention over visible voxels. This operation, for a query located at $\mathbf{x}$, is expressed as

$$\text{DSA}(\tilde{\mathbf{V}}_{\text{vis}}^{\mathbf{x}}, \tilde{\mathbf{V}}_{\text{vis}}, \mathbf{x}) = \sum_{n=1}^{N} A_n W \psi(\tilde{\mathbf{V}}_{\text{vis}}, \mathbf{x} + \Delta\mathbf{x}) . \tag{4}$$

Inspired by CGFormer´s [59] query generator, our instance generator transforms the DSA output to initialize instance proposals via

$$\mathbf{I}_P = \text{DSA}(\tilde{\mathbf{V}}_{\text{vis}}^{\mathbf{x}}, \tilde{\mathbf{V}}_{\text{vis}}, \mathbf{x}) + \mathbf{W_I} \in \mathbb{R}^{K \times C} , \tag{5}$$

where $\mathbf{W_I}$ represents learnable embeddings to improve the representational capacity, and $K$ denotes the maximum number of detectable instances. This initialization typically directs feature aggregation to highly relevant, instance-dependent image regions, as shown in Fig. 3.

**Voxel Proposals.** To initialize voxel proposals $\mathbf{V}_P$, we first merge the visible and invisible voxels through element-wise addition along dimension $C$, denoted as $\tilde{\mathbf{V}}_{\text{vis}} \oplus \mathbf{V}_{\text{invis}}$. We then apply 3D deformable self-attention to distribute updated context information over the entire scene volume, formulated as $\mathbf{V}_P = \text{DSA}(\tilde{\mathbf{V}}_{\text{vis}} \oplus \mathbf{V}_{\text{invis}})$, which is in line with [32, 21, 59]:

## 3.4 Encoding

The purpose of this part of our architecture is to (i) transform voxel proposals $\mathbf{V}_P$ into voxel features $\mathbf{V}_F$ that encode semantic information and (ii) project this semantic information onto the instance proposals $\mathbf{I}_P$, thus prompting semantic instance-voxel relationships.

To encode semantic information, we propagate the voxel proposals $\mathbf{V}_P$ through a 3D Local and Global Encoder (LGE) based on CGFormer [59]. This encoder enhances the semantic and geometric discriminability of the scene volume by integrating TPV [20] and voxel representations, effectively capturing global and local contextual features, respectively. For an in-depth discussion, we direct readers to [59]. The 3D LGE $\Phi_{\text{LGE}}$ transforms voxel proposals $\mathbf{V}_P$ into semantic voxel features $\mathbf{V}_F \in \mathbb{R}^{X \times Y \times Z \times C}$, formulated as $\mathbf{V}_F = \Phi_{\text{LGE}}(\mathbf{V}_P)$. Building on these semantically encoded voxel features, we apply cross-attention (CA) to instance proposals $\mathbf{I}_P \in \mathbb{R}^{K \times C}$ (as queries) and semantic voxel features $\mathbf{V}_F \in \mathbb{R}^{X \times Y \times Z \times C}$ (as keys and values), producing updated instance features $\tilde{\mathbf{I}}_P^{\text{CA}}$. Specifically, the updated instance features are computed as:

$$\tilde{\mathbf{I}}_P^{\text{CA}} = \text{CA}(\mathbf{I}_P, \mathbf{V}_F) = \sum_{n=1}^{N} A_n W \mathbf{V}_F(\mathbf{p}_n) , \tag{6}$$

where $N = X \cdot Y \cdot Z$ denotes the total number of voxels, $\mathbf{p}_n$ represents the $n$-th voxel position, $A_n \in [0, 1]$ is the attention weight, and $W$ is the learnable projection weight.

Subsequently, we apply self-attention (SA) to the updated instance features $\tilde{\mathbf{I}}_P^{\text{CA}}$ to capture global dependencies among the $K$ instances, yielding further refined instances $\tilde{\mathbf{I}}_P^{\text{SA}}$. The self-attention mechanism is defined as:

$$\tilde{\mathbf{I}}_P^{\text{SA}} = \text{SA}(\tilde{\mathbf{I}}_P^{\text{CA}}) = \sum_{n=1}^{K} A_n W \tilde{\mathbf{I}}_P^{\text{CA}}(n) , \tag{7}$$

where $\tilde{\mathbf{I}}_P^{\text{CA}}(n)$ denotes the feature vector of the $n$-th instance from $\tilde{\mathbf{I}}_P^{\text{CA}}$. To obtain the final encoded semantic instance features, we further process the self-attention output $\tilde{\mathbf{I}}_P^{\text{SA}}$ through an MLP Encoder $\Phi_E$, yielding $\mathbf{I}_F \in \mathbb{R}^{K \times C}$. This step refines the instance representations for downstream tasks. The encoding is computed as $\mathbf{I}_F = \Phi_E(\tilde{\mathbf{I}}_P^{\text{SA}}) \in \mathbb{R}^{K \times C}$.

## 3.5 Decoding

Since we employ a two-stage training scheme (see Sec. 3.6), the goal of decoding is twofold. In the first stage, a lightweight MLP head $\phi_{\text{SSC}}$ predicts the semantic scene via $\phi_{\text{SSC}}(\mathbf{V}_F)$.

In the second training stage, a lightweight panoptic head decodes instance features $\mathbf{I}_F$ to semantic class logits $\mathbf{I}_L \in \mathbb{R}^{K \times L}$, where $K$ is the number of instances and $L$ is the number of semantic classes. These logits are further processed with voxel features $\mathbf{V}_F$ to perform instance-voxel alignment that eventually enables panoptic scene aggregation, inspired by PaSCo [3]. To elaborate, we align the instance features $\mathbf{I}_F$ with the voxel features $\mathbf{V}_F \in \mathbb{R}^{X \times Y \times Z \times C}$ to compute affinity scores $\mathbf{H} \in \mathbb{R}^{X \times Y \times Z \times K}$ by applying the dot product, resulting in $\mathbf{H} = \mathbf{V}_F \cdot \mathbf{I}_F^{\top}$. These scores are converted to predicted probabilities via $\mathbf{H}_P = \text{sigmoid}(\mathbf{H})$, where $\mathbf{H}_P$ denotes the model's confidence in each voxel belonging to each instance. For panoptic aggregation [3], voxel-wise logits

$\mathbf{V}_L \in \mathbb{R}^{X \times Y \times Z \times L}$ are computed as $\mathbf{V}_L = \mathbf{H}_P \cdot \mathbf{I}_L$. It should be emphasized that voxel-wise logits do not represent the probability of belonging to a certain semantic class, but the confidence of belonging to one of the instances. Furthermore, semantic class IDs are assigned via argmax over $\mathbf{V}_L$ along $L$, and instance IDs via argmax over $\mathbf{H}_P^\top$ along $K$, yielding a complete panoptic scene. Note that the alignment process and the panoptic aggregation do not include learnable parameters.

## 3.6 Training Objective

We follow a two-stage training strategy in which we formulate the learning objectives for SSC and PSC tasks sequentially. Technically, the **first stage** optimizes

$$\mathcal{L}_{\text{SSC}} = \lambda_{\text{ce}} \mathcal{L}_{\text{ce}} + \lambda_{\text{depth}} \mathcal{L}_{\text{depth}} + \lambda_{\text{sem}} \mathcal{L}_{\text{scal}}^{\text{sem}} + \lambda_{\text{geo}} \mathcal{L}_{\text{scal}}^{\text{geo}} \,, \tag{8}$$

where the weights for the cross-entropy loss, semantic scene-class affinity loss (SCAL), and geometric SCAL [4] are set to $\lambda_{\text{ce}} = \lambda_{\text{sem}} = \lambda_{\text{geo}} = 1$, respectively, and the weight for the depth loss [59] is set to $\lambda_{\text{depth}} = 10^{-4}$.

The **second stage** calculates losses mask-wise by comparing each ground-truth mask with the best matching predicted mask. Consistent with previous research [3, 47], we find the best match by applying the Hungarian method [26] to perform bipartite matching [5], using an IoU threshold of $> 50\%$. This threshold follows the standard convention established in prior work on panoptic segmentation [23, 3], and the Hungarian algorithm ensures a globally optimal, permutation-invariant one-to-one assignment. After matching, the second stage optimizes

$$\mathcal{L}_{\text{PSC}} = \lambda_{\text{mask}} \mathcal{L}_{\text{ce}}^{\text{mask}} + \lambda_{\text{dice}} \mathcal{L}_{\text{dice}} + \lambda_{\text{focal}} \mathcal{L}_{\text{focal}} + \lambda_{\text{depth}} \mathcal{L}_{\text{depth}} \,, \tag{9}$$

where $\mathcal{L}_{\text{ce}}^{\text{mask}}$ represents mask-wise cross-entropy loss, $\mathcal{L}_{dice}$ is the dice loss of [37] and $\mathcal{L}_{focal}$ is the focal loss based on [33]. In practice, we set the weights of these losses to $\lambda_{\text{mask}} = \lambda_{\text{dice}} = 1$ and $\lambda_{\text{focal}} = 40$, respectively, following PaSco [3]. Moreover, $\mathcal{L}_{\text{depth}}$ is defined identically to that in $\mathcal{L}_{\text{SSC}}$.

## 4 Experiments

### 4.1 Quantitative Results

**Dataset and Baselines.** We conduct our experiments by (i) in-domain training and testing on the SemanticKITTI SSC dataset [1], and (ii) out-of-domain zero-shot generalization on the distinct SSCBench-KITTI360 [31]. The instance ground-truths for both datasets are provided by PaSco [3]. Given the novelty of vision-based PSC, no established baselines currently exist for direct comparison. Furthermore, recall from Sec. 2 that PanoSSC [47] represents an existing vision-based PSC method using forward-facing imagery. This would make it a suitable baseline for comparison. However, this approach is trained on a non-public post-processed version of the SemanticKITTI dataset [1], which, in addition to the absence of released source code, challenges direct comparison with our method (see Sec. A.6 of the technical appendix for further details). Consequently, we adapt the latest state-of-the-art vision-based SSC methods [4, 21, 63, 59] and generate instance predictions by clustering Thing-classes from their outputs, in line with PaSco [3], which introduced the task of 3D PSC. In particular, to ensure fairness and methodological rigor, we apply DBSCAN [13], the same Euclidean clustering algorithm used to construct the PSC ground truth in PaSco. Furthermore, we utilize pre-trained checkpoints for the baselines, acquired from the official publicly available implementations. We provide further details in Sec. A.1 of the technical appendix.

**In-Domain Performance.** In summary, IPFormer exceeds all baselines in overall panoptic metrics PQ and PQ$^\dagger$, and achieves best or second-best results on individual metrics, as shown in Tab. 1. These results showcase the significant advancements IPFormer brings to Panoptic Scene Completion, highlighting its robustness and efficiency. Moreover, IPFormer attains superior performance on RQ-All, while securing the second-best position in SQ-All and PQ-Thing. Furthermore, although our approach places second in SQ-Stuff, it surpasses existing methods in RQ-Stuff and notably PQ-Stuff, indicating superior capability in recognizing Stuff-classes accurately. Notably, despite our method achieving state-of-the-art performance in PSC, it exhibits moderate SSC performance on in-domain data. These results are evaluated on the PSC output of our two-staged, fully-trained model. For SSC metrics, the instance IDs of all voxels are disregarded, effectively reducing PSC to SSC. Additionally, our method directly predicts a full panoptic scene, resulting in a significantly superior runtime of

Table 1: In-domain performance on SemanticKITTI val. set [1]. Best and second-best results are bold and underlined, respectively. Due to the absence of established baselines for vision-based PSC (see Sec. 4.1), we infer state-of-the-art SSC methods and apply DBSCAN [13] to retrieve instances.

| | | | | | PSC Metrics | | | | | | SSC Metrics | | |
| | | All | | | | Thing | | | Stuff | | | | |
| Method | PQ†↑ | PQ↑ | SQ↑ | RQ↑ | PQ↑ | SQ↑ | RQ↑ | PQ↑ | SQ↑ | RQ↑ | IoU↑ | mIoU↑ | Runtime [s] ↓ |
|---|---|---|---|---|---|---|---|---|---|---|---|---|---|
| MonoScene [4] + DBSCAN | 10.12 | 3.43 | 15.15 | 5.33 | 0.51 | 7.36 | 0.87 | 5.56 | 20.81 | 8.57 | 36.80 | 11.31 | 4.51 |
| Symphonies [21] + DBSCAN | 11.69 | 3.75 | 26.09 | 5.95 | 1.07 | 27.65 | 1.76 | 5.70 | 24.95 | 8.99 | 41.92 | 15.02 | 4.54 |
| OccFormer [63] + DBSCAN | 11.25 | 4.32 | 24.19 | 6.69 | 0.68 | 21.47 | 1.15 | 6.96 | 26.16 | 10.73 | 36.43 | 13.51 | 4.70 |
| CGFormer [59] + DBSCAN | 14.39 | 6.16 | **48.14** | 9.48 | **2.20** | **44.46** | **3.47** | 9.03 | **50.82** | 13.86 | **45.98** | **16.89** | 4.70 |
| IPFormer (ours) | **14.45** | **6.30** | 41.95 | **9.75** | 2.09 | 42.67 | 3.33 | **9.35** | 41.43 | **14.43** | 40.90 | 15.33 | **0.33** |

Table 2: Out-of-domain zero-shot generalization performance of IPFormer and the closest baseline CGFormer+DBSCAN, by training on SemanticKITTI [1] and cross-validating on SSCBench-KITTI360 test set [31]. IPFormer demonstrates superior absolute and relative generalization performance across PSC and SSC metrics.

| | | | | | PSC Metrics | | | | | | SSC Metrics | |
| | | All | | | | Thing | | | Stuff | | | | |
| | PQ†↑ | PQ↑ | SQ↑ | RQ↑ | PQ↑ | SQ↑ | RQ↑ | PQ↑ | SQ↑ | RQ↑ | IoU↑ | mIoU↑ |
|---|---|---|---|---|---|---|---|---|---|---|---|---|
| **SemanticKITTI** | | | | | | | | | | | | |
| CGFormer [59] + DBSCAN | 14.39 | 6.16 | **48.14** | 9.48 | **2.20** | **44.46** | **3.47** | 9.03 | **50.82** | 13.86 | **45.98** | **16.89** |
| IPFormer (ours) | **14.45** | **6.30** | 41.95 | **9.75** | 2.09 | 42.67 | 3.33 | **9.35** | 41.43 | **14.43** | 40.90 | 15.33 |
| **KITTI-360** | | | | | | | | | | | | |
| CGFormer [59] + DBSCAN | 8.44 | 1.08 | 17.82 | 1.87 | **0.53** | 20.06 | **0.96** | 1.48 | 16.19 | 2.54 | 28.11 | 9.44 |
| IPFormer (ours) | **9.41** | **1.23** | **24.68** | **2.16** | 0.52 | **22.76** | 0.95 | **1.68** | **25.89** | **2.93** | **28.74** | **9.53** |
| **Relative Gap ↓** | | | | | | | | | | | | |
| CGFormer [59] + DBSCAN | 41.37% | 82.47% | 62.98% | 80.28% | 75.91% | 54.89% | 72.34% | 83.61% | 68.15% | 81.67% | 38.88% | 44.09% |
| IPFormer (ours) | **34.88%** | **80.48%** | **41.19%** | **77.85%** | **75.12%** | **46.64%** | **71.53%** | **82.03%** | **37.52%** | **79.69%** | **29.73%** | **37.81%** |

0.33 s, thus providing a runtime reduction of over 14×. In the technical appendix, we further report class-wise quantitative results and a detailed runtime analysis, including memory utilization during training.

**Out-of-Domain Zero-Shot Generalization Performance.** Trained on SemanticKITTI [1], we cross-validate our method on the distinct SSCBench-KITTI360 dataset [31]. Results in Tab. 2 show that IPFormer demonstrates superior zero-shot generalization capability. Especially in comparison to its closest baseline, CGFormer+DBSCAN, which it outperforms across PSC and SSC metrics. These results highlight the generalization capability of IPFormer by leveraging out-of-domain image context to robustly initialize context-adaptive instance proposals, resulting in superior SSC and PSC performance.

## 4.2 Ablation Study

**Context-Adaptive Initialization.** We compare our introduced context-adaptive instance proposals with context-aware, but non-adaptive instance queries (Tab. 3). While initializing proposals adaptively slightly decreases SQ-Stuff, it remarkably increases all Thing-related metrics by $18.65\%$ on average, with RQ-Things showing the best performance gain of $21.98\%$. This insight demonstrates that adaptively initializing instance proposals from image context not only enables us to drastically recognize more objects, but also enables Thing segmentation with substantially higher quality.

**Visibility-Based Sampling Strategy.** We examine our proposed visible-only approach for proposal initialization to a method that employs both visible and invisible voxels (Tab. 4). Formally, these proposals are initialized similar to Eq. (5), specifically as $\text{DSA}(\tilde{\mathbf{V}}^{\mathbf{x}}_{\text{vis}} \oplus \mathbf{V}^{\mathbf{x}}_{\text{invis}}, \tilde{\mathbf{V}}_{\text{vis}} \oplus \mathbf{V}_{\text{invis}}, \mathbf{x}) + \mathbf{W_I}$. Under our visible-only approach, almost all metrics increase significantly, most notably RQ-Thing and SQ-Thing by $48.00\%$ and $93.95\%$, respectively, thus evidently enhancing recognition and segmentation of objects from the Thing-category. These findings confirm that visible voxels carry a robust reconstruction signal, which can be leveraged to initialize instance proposals and eventually improve PSC performance.

**Deep Supervision.** Inspired by PanoSSC [47], we investigate deep supervision in the form of auxiliary losses, essentially supervising the attention maps of each layer during the encoding of the instance proposals (Tab. 5). Our findings show that guiding intermediate layers in such way degrades all metrics. Most notably, when deep supervision is not applied, SQ-Thing registers a substantial improvement of $102.61\%$.

Table 3: Ablation on the initialization of instance queries vs. instance proposals.

| Method | instance queries | instance proposals |
|---|---|---|
| **All** | | |
| PQ†↑ | **14.80** | 14.45 (−2.37 %) |
| PQ↑ | 6.08 | **6.30** (+3.62 %) |
| SQ↑ | **42.65** | 41.95 (−1.64 %) |
| RQ↑ | 9.36 | **9.75** (+4.17 %) |
| **Thing** | | |
| PQ↑ | 1.76 | **2.09** (+18.75 %) |
| SQ↑ | 37.07 | **42.67** (+15.23 %) |
| RQ↑ | 2.73 | **3.33** (+21.98 %) |
| **Stuff** | | |
| PQ↑ | 9.22 | **9.35** (+1.41 %) |
| SQ↑ | **46.71** | 41.43 (−11.30 %) |
| RQ↑ | 14.18 | **14.43** (+1.76 %) |

Table 4: Ablation on the visibility-based sampling strategy for proposal initialization.

| Method | visible + invisible | visible only |
|---|---|---|
| **All** | | |
| PQ†↑ | **14.85** | 14.45 (−2.69 %) |
| PQ↑ | 5.86 | **6.30** (+7.51 %) |
| SQ↑ | 30.54 | **41.95** (+37.36 %) |
| RQ↑ | 9.01 | **9.75** (+8.21 %) |
| **Thing** | | |
| PQ↑ | 1.40 | **2.09** (+49.28 %) |
| SQ↑ | 22.00 | **42.67** (+93.95 %) |
| RQ↑ | 2.25 | **3.33** (+48.00 %) |
| **Stuff** | | |
| PQ↑ | 9.11 | **9.35** (+2.63 %) |
| SQ↑ | 36.74 | **41.43** (+12.77 %) |
| RQ↑ | 13.93 | **14.43** (+3.59 %) |

Table 5: Ablation on deep supervision for instance encoding during second-stage training.

| Method | w/ deep supervision | w/o deep supervision |
|---|---|---|
| **All** | | |
| PQ†↑ | 14.36 | **14.45** (+0.63 %) |
| PQ↑ | 5.77 | **6.30** (+9.19 %) |
| SQ↑ | 32.51 | **41.95** (+29.04 %) |
| RQ↑ | 9.06 | **9.75** (+7.62 %) |
| **Thing** | | |
| PQ↑ | 1.26 | **2.09** (+65.87 %) |
| SQ↑ | 21.06 | **42.67** (+102.61 %) |
| RQ↑ | 2.00 | **3.33** (+66.50 %) |
| **Stuff** | | |
| PQ↑ | 9.06 | **9.35** (+3.20 %) |
| SQ↑ | 40.83 | **41.43** (+1.47 %) |
| RQ↑ | 14.20 | **14.43** (+1.62 %) |

Table 6: Ablation on the dual-head design and the training strategy. Methods (a)-(d) evaluate combinations of single/dual-head and one/two-stage trainings, where $*$ denotes frozen weights of the first stage during second-stage training. Methods (e)-(i) examine the incorporation of the first-stage loss $\mathcal{L}_{SSC}$ by different factors of $\lambda_{SSC}$ into the second stage. (j) represents the final IPFormer config.

| Method | Dual Head | Two Stage | $\lambda_{SSC}$ | All **PQ†↑** | **PQ↑** | SQ↑ | RQ↑ | Thing **PQ↑** | SQ↑ | RQ↑ | Stuff **PQ↑** | SQ↑ | RQ↑ |
|---|---|---|---|---|---|---|---|---|---|---|---|---|---|
| (a) | | | - | 14.64 | 6.23 | 33.37 | 9.54 | 1.75 | 15.40 | 2.66 | **9.49** | **46.44** | **14.55** |
| (b) | ✓ | | - | 14.21 | 5.57 | 36.09 | 8.58 | 1.06 | 22.29 | 1.73 | 8.84 | 46.12 | 13.57 |
| (c) | | ✓ | - | 10.66 | 0.42 | 14.31 | 0.76 | 0.13 | 7.07 | 0.22 | 0.63 | 19.57 | 1.15 |
| (d) | ✓* | ✓ | - | 13.75 | 5.06 | 38.15 | 7.98 | 0.27 | 27.31 | 0.49 | 8.55 | 46.00 | 13.43 |
| (e) | ✓ | ✓ | 1.00 | 14.35 | 6.06 | 38.74 | 9.30 | 1.66 | 35.52 | 2.63 | 9.27 | 41.08 | 14.16 |
| (f) | ✓ | ✓ | 0.50 | 14.42 | 6.27 | 38.72 | 9.69 | 1.88 | 42.01 | 3.13 | 9.46 | 36.33 | **14.45** |
| (g) | ✓ | ✓ | 0.20 | 14.46 | 6.05 | 33.06 | 9.37 | 1.60 | 28.79 | 2.54 | 9.29 | 36.18 | 14.34 |
| (h) | ✓ | ✓ | 0.10 | 14.41 | 6.08 | 33.50 | 9.40 | 1.72 | 29.93 | 2.73 | 9.26 | 36.27 | 14.26 |
| (i) | ✓ | ✓ | 0.01 | **14.67** | 5.16 | 24.33 | 7.97 | 0.90 | 7.83 | 1.43 | 8.27 | 36.32 | 12.73 |
| (j) | ✓ | ✓ | - | 14.45 | **6.30** | **41.95** | **9.75** | **2.09** | **42.67** | **3.33** | 9.35 | 41.43 | 14.43 |

**Dual-head and Two-Stage Training.** The baseline (a) in Tab. 6 uses a single head and single-stage training, performing well on Stuff but poorly in SQ-Thing. Introducing separate heads for SSC and PSC in (b) slightly reduces Stuff performance but improves SQ-Thing to nearly match the baseline. Replacing the dual-head with a purely two-stage approach in (c) severely degrades SQ and yields the worst RQ scores. In contrast, adopting both the dual-head architecture and two-stage training in (j) achieves the best overall results. Alternatively, freezing the first stage during the second-stage training (d) proves detrimental, notably reducing RQ-Thing. Furthermore, methods (e)-(i) evaluate the integration of the SSC objective into the second-stage training, by adding $\mathcal{L}_{SSC}$ to $\mathcal{L}_{PSC}$ via different factors of $\lambda_{SSC}$. Overall, these configurations fall short of the superior performance achieved by our final method (j), which applies both a dual-head architecture and a two-stage training strategy that separates SSC and PSC objectives. Note that these findings are underscored by additional ablation experiments presented in Sec. A.7 of the technical appendix.

## 4.3 Qualitative Results

Presented in Tab. 4, IPFormer surpasses existing approaches by excelling at identifying individual instances, inferring their semantics, and reconstructing geometry with exceptional fidelity. Even for extremely low-frequency categories such as the person category (0.07 %) under adverse lighting conditions, and in the presence of trace-artifacts from dynamic objects in the ground-truth data, our method proves visually superior. These advancements stem from IPFormer's instance proposals, which dynamically adapt to scene characteristics, thus preserving high precision in instance identification, semantic segmentation, and geometric completion. Conversely, other models tend to encounter challenges in identifying semantic instances effectively while simultaneously retaining geometric integrity. Moreover, our **instance-specific saliency analysis** in Fig. 3 underscores these findings.

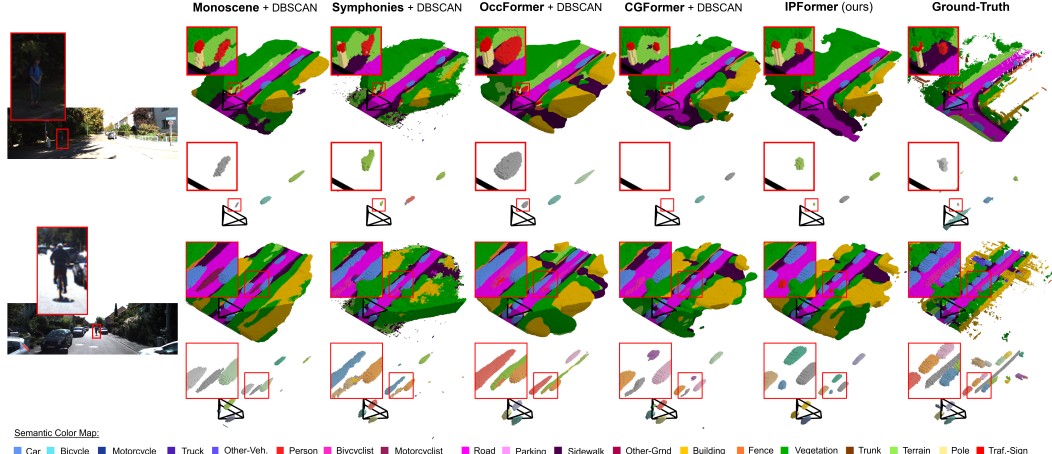

Figure 4: Qualitative results on the SemanticKITTI val. set [1]. Each top row illustrates purely semantic information, following the SSC color map. Each bottom row displays individual instances, with randomly assigned colors to facilitate differentiation. Note that we specifically show instances of the Thing-category for clarity.

## 5   Conclusion

IPFormer advances the field of 3D Panoptic Scene Completion by leveraging context-adaptive instance proposals derived from camera images at both train and test time. Its contributions are reflected in achieving state-of-the-art in-domain performance, exhibiting superior zero-shot generalization on out-of-domain data, and achieving a runtime reduction exceeding $14\times$. Ablation studies confirm the critical role of visibility-based proposal initialization, the dual-head architecture and the two-stage training strategy, while qualitative results underscore the method's ability to reconstitute true scene geometry despite incomplete or imperfect ground truth. Taken together, these findings serve as a promising foundation for downstream applications like autonomous driving and future research in holistic 3D scene understanding.

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

# A  Technical Appendix

## A.1  Experimental setup

**Datasets.**    We utilize the SemanticKITTI dataset, a large-scale urban dataset designed for Semantic Scene Completion. SemanticKITTI provides 64-layer LiDAR scans voxelized into grids of $256{\times}256{\times}32$ with 0.2m voxel resolution, alongside RGB images of $1226{\times}370$ pixel resolution, covering 20 distinct semantic classes (19 labeled classes plus 1 free class). The dataset comprises 10 training sequences, 1 validation sequence, and 11 test sequences, with our experiments adhering to the standard split [44] of 3834 training and 815 validation grids.

To enable **panoptic evaluation**, we adopt the PaSCo dataset [3], which extends by generating pseudo panoptic labels. PaSCo employs DBSCAN [13] to cluster voxels of "thing" classes into distinct instance IDs, using a distance threshold of $\epsilon = 1$ and a minimum group size of MinPts $= 8$. The authors of PaSCo ensure that the pseudo labels are valid by comparing them against the available LiDAR single-scan point-wise panoptic ground truth from the validation set. Both the pseudo labels (generated by DBSCAN) and the ground truth are voxelized, and their quality is assessed in regions where both are defined. For more details, including quantitative and qualitative evaluation, see PaSco [3, supplementary material Sec. 8.2]. Since instance labels cannot be derived for SemanticKITTI's hidden test set, we perform evaluations on the validation set.

**Metrics.**    To evaluate our Panoptic Scene Completion approach, we employ the Panoptic Quality (PQ) metric [24], which combines Segmentation Quality (SQ) and Recognition Quality (RQ). PQ is defined as:

$$\text{PQ} = \text{SQ} \times \text{RQ} = \frac{\sum_{(p,g)\in TP} \text{IoU}(p,g)}{|TP| + \frac{1}{2}|FP| + \frac{1}{2}|FN|} \ , \tag{10}$$

where $\text{SQ} = \frac{\sum_{(p,g)\in TP} \text{IoU}(p,g)}{|TP|}$ and $\text{RQ} = \frac{|TP|}{|TP| + \frac{1}{2}|FP| + \frac{1}{2}|FN|}$. $TP$, $FP$, and $FN$ represent true positives, false positives, and false negatives, respectively, and IoU is the intersection-over-union. SQ measures segmentation fidelity via the average IoU of matched segments, while RQ assesses recognition accuracy as the F1-score [10]. We compute PQ, SQ, and RQ across all classes, as well as separately for "stuff" (amorphous regions) and "things" (countable objects), to analyze category-specific performance.

The standard PQ metric requires a predicted segment to match a ground-truth segment with IoU $> 0.5$. However, this strict threshold can be overly conservative for Stuff classes, which typically lack well-defined boundaries. The metric $\text{PQ}^{\dagger}$ [42] relaxes the matching criterion specifically for Stuff classes. Formally, $\text{PQ}^{\dagger}$ retains the same formulation as PQ:

$$\text{PQ}^{\dagger} = \frac{\sum_{(p,g)\in \text{TP}^{\dagger}} \text{IoU}(p,g)}{|\text{TP}^{\dagger}| + \frac{1}{2}|\text{FP}^{\dagger}| + \frac{1}{2}|\text{FN}^{\dagger}|}, \tag{11}$$

but relaxes the matching condition used to define true positives ($\text{TP}^{\dagger}$), and thus false positives ($\text{FP}^{\dagger}$) and false negatives ($\text{FN}^{\dagger}$). Specifically, for Thing classes, predicted and ground-truth segment pairs $(p,g)$ are matched if $\text{IoU}(p,g) > 0.5$, identical to the original PQ definition. For Stuff classes, matches are accepted if $\text{IoU}(p,g) > 0$, thereby allowing any overlapping prediction to contribute to the metric. This relaxation acknowledges the inherent ambiguity in delineating stuff regions and reduces penalties for minor misalignments. As with PQ, we compute $\text{PQ}^{\dagger}$ jointly across all classes and separately for stuff and thing categories to enable detailed performance analysis.

## A.2  Implementation Details

**Training and Architecture.**    In accordance with [4, 20, 32, 21], we train for 25 epochs in the first stage and 30 epochs in the second stage, using AdamW [36] optimizer with standard hyperparameters $\beta_1 = 0.9$, $\beta_2 = 0.99$, and a batch size of 1. We utilize a single NVIDIA A100 80GB GPU, adopt a maximum learning rate of $1 \times 10^{-4}$, and implement a cosine adaptive learning rate schedule decay, with a cosine warmup applied over the initial 2 epochs. Our implementation is based on PyTorch [39]

Table 7: Class-wise quantitative results on SemanticKITTI val. set [1] (**best**, second-best) with corresponding class frequencies. The asterisk ($*$) indicates SSC methods, for which the outputs are clustered to identify their instances, as described in Sec. 4.1.

| | Method | Car (3.92%) | Bicycle (0.03%) | Motorcycle (0.03%) | Truck (0.16%) | Other-Vehicle (0.20%) | Person (0.07%) | Bicyclist (0.07%) | Motorcyclist. (0.05%) | Road (15.30%) | Parking (1.12%) | Sidewalk (11.13%) | Other-Ground (0.56%) | Building (14.10%) | Fence (3.90%) | Vegetation (39.30%) | Trunk (0.51%) | Terrain (9.17%) | Pole (0.29%) | Traffic-Sign (0.08%) | Mean |
|---|---|---|---|---|---|---|---|---|---|---|---|---|---|---|---|---|---|---|---|---|---|
| **PQ** | MonoScene [4]* | 4.10 | 0.00 | 0.00 | 0.00 | 0.00 | 0.00 | 0.00 | 0.00 | 53.92 | 1.27 | 1.01 | 0.00 | 0.00 | 0.00 | 0.00 | 0.00 | 4.95 | 0.00 | 0.00 | 3.43 |
| | Symphonies [21]* | 7.80 | 0.00 | 0.00 | 0.14 | 0.37 | 0.22 | 0.00 | 0.00 | 54.79 | 0.46 | 1.92 | 0.00 | 0.00 | 0.00 | 0.63 | 0.00 | 4.88 | 0.00 | 0.00 | 3.75 |
| | OccFormer [63]* | 3.51 | 0.00 | 0.00 | 1.60 | 0.33 | 0.00 | 0.00 | 0.00 | 59.52 | 3.49 | 6.79 | 0.00 | 0.00 | 0.00 | 0.51 | 0.00 | 6.30 | 0.00 | 0.00 | 4.32 |
| | CGFormer [59]* | **14.14** | **0.58** | **1.14** | 0.99 | 0.51 | 0.25 | 0.00 | 0.00 | **66.78** | 3.43 | 11.08 | 0.00 | **0.48** | **0.09** | 1.14 | **0.10** | **15.35** | **0.63** | **0.28** | 6.16 |
| | IPFormer (ours) | 12.83 | 0.45 | 0.56 | **1.69** | **0.92** | **0.27** | 0.00 | 0.00 | 66.28 | **6.16** | **14.52** | 0.00 | 0.13 | 0.00 | **2.28** | 0.00 | 13.25 | 0.12 | 0.15 | **6.30** |
| **SQ** | Monoscene [4]* | 58.89 | 0.00 | 0.00 | 0.00 | 0.00 | 0.00 | 0.00 | 0.00 | 66.55 | 55.27 | 52.11 | 0.00 | 0.00 | 0.00 | 0.00 | 0.00 | 55.00 | 0.00 | 0.00 | 15.15 |
| | Symphonies [21]* | 61.45 | 0.00 | 0.00 | 51.35 | 52.88 | 55.47 | 0.00 | 0.00 | 65.11 | 50.71 | 53.09 | 0.00 | 0.00 | 0.00 | 51.64 | 0.00 | 53.94 | 0.00 | 0.00 | 26.09 |
| | OccFormer [63]* | 58.62 | 0.00 | 0.00 | **61.53** | 51.59 | 0.00 | 0.00 | 0.00 | 68.26 | 56.29 | 54.69 | 0.00 | 0.00 | 0.00 | 52.22 | 0.00 | 56.33 | 0.00 | 0.00 | 24.19 |
| | CGFormer [59]* | 65.59 | **52.01** | 54.16 | 57.16 | **59.22** | **67.53** | 0.00 | 0.00 | 70.37 | 54.51 | 55.15 | 0.00 | 50.91 | **52.86** | 52.41 | **51.43** | 58.94 | **58.72** | **53.72** | **48.14** |
| | IPFormer (ours) | **65.84** | 51.25 | **58.60** | 52.34 | 57.34 | 56.02 | 0.00 | 0.00 | **70.40** | **56.44** | **55.36** | 0.00 | **52.93** | 0.00 | **52.68** | 0.00 | **59.37** | 57.59 | 50.91 | 41.95 |
| **RQ** | MonoScene [4]* | 6.97 | 0.00 | 0.00 | 0.00 | 0.00 | 0.00 | 0.00 | 0.00 | 81.02 | 2.30 | 1.94 | 0.00 | 0.00 | 0.00 | 0.00 | 0.00 | 9.01 | 0.00 | 0.00 | 5.33 |
| | Symphonies[21]* | 12.69 | 0.00 | 0.00 | 0.27 | 0.70 | 0.40 | 0.00 | 0.00 | 84.15 | 0.91 | 3.61 | 0.00 | 0.00 | 0.00 | 1.22 | 0.00 | 9.04 | 0.00 | 0.00 | 5.95 |
| | OccFormer[63]* | 5.98 | 0.00 | 0.00 | 2.60 | 0.64 | 0.00 | 0.00 | 0.00 | 87.20 | 6.20 | 12.41 | 0.00 | 0.00 | 0.00 | 0.98 | 0.00 | 11.19 | 0.00 | 0.00 | 6.69 |
| | CGFormer[59]* | **21.56** | **1.11** | **2.11** | 1.73 | 0.87 | 0.37 | 0.00 | 0.00 | **94.91** | 6.30 | 20.09 | 0.00 | **0.94** | **0.17** | 2.18 | **0.20** | **26.05** | **1.06** | **0.51** | 9.48 |
| | IPFormer (ours) | 19.49 | 0.88 | 0.95 | **3.23** | **1.60** | **0.49** | 0.00 | 0.00 | 94.16 | **10.91** | **26.23** | 0.00 | 0.24 | 0.00 | **4.32** | 0.00 | 22.31 | 0.22 | 0.30 | **9.75** |

with an fp32 backend. Moreover, we operate on a $50\%$ voxel grid resolution of $X = 128$, $Y = 128$, $Z = 16$ and finally upsample to the ground-truth grid resolution of $256 \times 256 \times 32$ via trilinear interpolation. The feature dimension is set to $C = 128$. Training IPFormer takes approximately 3.5 days for each of the two stages. The second stage training is initialized with the final model state of the first stage, and we eventually present results for the best checkpoint based on $PQ^\dagger$. Aligning with [32, 21, 59], we adopt a pretrained MobileStereoNet [46] to estimate depth maps, and employ EfficientNetB7 [49] as our image backbone, consistent with [63, 59]. Moreover, the context net consists of a lightweight CNN, while the panoptic head represents a single linear layer for projection to class logits. The deformable cross and self attention blocks during proposal initialization consist of three layers and two layers, respectively, while 8 points are sampled for each reference point. Finally, the cross and self-attention blocks during decoding each consist of three layers.

**Clustering.** To cluster the predictions of SSC baselines and retrieve individual instances, we apply DBSCAN [13] with parameters $\epsilon = 1$ and MinPts=8, in line with the work of PaSCo [3], which provides ground-truth instances for the SemanticKITTI dataset [1]. The clustering is performed on an AMD EPYC 7713 CPU (allocating 16 cores) with 64GB memory.

## A.3 Class-Wise Quantitative Results

In addition to the overall performance on Panoptic Scene Completion in Tab. 1, we report class-wise results in Tab. 7. IPFormer consistently ranks first or second, thereby demonstrating state-of-the-art performance in vision-based PSC, aligning with our primary findings.

As also shown in Tab. 1, all methods showcase suboptimal performance on Thing classes, which arises from the significant class imbalance in the SemanticKITTI dataset, where Thing classes make up only $4.53\%$ of all voxels. To address this, we employ the Sigmoid Focal Loss (Eq. 12), which down-weights easier examples and focuses on harder-to-classify ones, particularly rare Thing classes. Sec. A.7 presents and discusses additional ablation results on the Focal Loss to demonstrate its effectiveness. Our proposed method balances performance between Thing and Stuff classes, achieving state-of-the-art results by prioritizing equitable performance across both categories rather than solely optimizing for Thing classes.

## A.4 Additional Qualitative Results.

We provide further qualitative results in Fig. 5, aligning with the primary results in that IPFormer reconstructs and identifies diverse objects of various sizes.

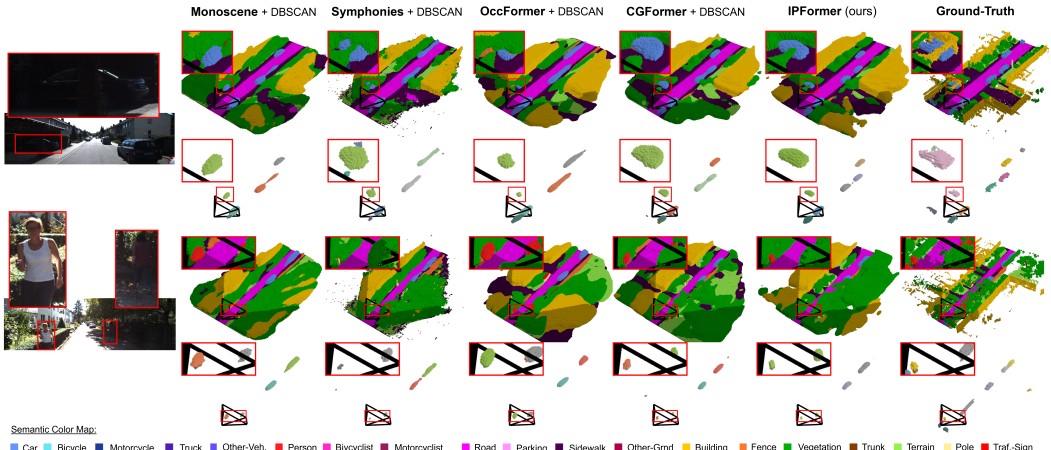

Figure 5: Additional qualitative results on the SemanticKITTI val. set [1]. Each top row illustrates purely semantic information, following the SSC color map. Each bottom row displays individual instances, with randomly assigned colors to facilitate differentiation. Note that we specifically show instances of the Thing category for clarity.

Table 8: Comparison of compute resources during training, and detailed runtime analysis. Additionally, we show the PQ$^\dagger$ metric on SemanticKITTI val. set [1] (**best**, second-best). Operations are performed on a single NVIDIA A100 GPU with $80\,$GB memory and an AMD EPYC 7713 CPU (allocating 16 cores) with $64\,$GB memory. The asterisk ($*$) indicates SSC methods, for which the outputs are clustered to identify their instances, as described in Sec. 4.1.

| Method | MonoScene [4]* | Symphonies [21]* | OccFormer [63]* | CGFormer [59]* | IPFormer (ours) |
|---|---|---|---|---|---|
| Inference Time [s]↓ | **0.08** | 0.11 | 0.27 | 0.27 | 0.33 |
| Clustering Time [s]↓ | 4.43 | 4.43 | 4.43 | 4.43 | **0.00** |
| Total Runtime [s]↓ | 4.51 | 4.54 | 4.70 | 4.70 | **0.33** |
| Training Memory [GB]↓ | 18.20 | 20.50 | **17.30** | 19.10 | 52.80 |
| PQ$^\dagger$↑ | 10.12 | 11.69 | 11.25 | 14.39 | **14.45** |

## A.5  Compute Resources and Runtime

In Tab. 8, we provide memory utilization during training and a detailed runtime analysis, in combination with the resulting performance for PQ$^\dagger$. Our method has the highest memory utilization with $52.8\,$GB, while OccFormer has the lowest with $17.3\,$GB. As elaborated in Sec. 4.1, we retrieve PSC predictions for the baselines by clustering their SSC predictions. Consequently, the total runtime for all baselines consists of inference time in addition to clustering time, with the latter being a constant of $4.43\,$s seconds for all baselines. Since our method directly predicts a panoptic scene, the total runtime is equal to the inference time. Thus, IPFormer exhibits a significantly superior runtime of $0.33\,$s compared to the second-best method in terms of PQ$^\dagger$, CGFormer, which has a total runtime of $4.70\,$s. IPFormer therefore provides a runtime reduction of over $14\times$.

## A.6  Limitations

**Experimental Results.** Experimental quantitative and qualitative results show IPFormer's state-of-the-art performance in vision-based PSC. However, there are remaining Thing-classes (*e.g.* Motorcyclist) and Stuff-classes (*e.g.* Other-Ground) which have not been recognized, due to their low class frequency or geometric fidelity. Despite these challenges, we believe that IPFormer's introduction of context-aware instance proposals will play a significant role in progressing 3D computer vision and specifically Panoptic Scene Completion.

**Comparison with PanoSSC.** As described in Sec. 4.1, we are not able to compare the performance of IPFormer with the vision-based PSC approach of PanoSSC [47], as this method is trained on a post-processed version of SemanticKITTI that is not publicly available. We aim to collaborate with the authors of PanoSSC to train IPFormer on this dataset and evaluate it on PanoSSC´s relaxed

Table 9: Ablation on IPFormer's PSC and SSC performance under PanoSSC's relaxed 20% IoU matching threshold on the SemanticKITTI dataset.

| IoU Threshold | PQ$^\dagger$ | PQ-All | SQ-All | RQ-All | PQ-Thing | SQ-Thing | RQ-Thing | PQ-Stuff | SQ-Stuff | RQ-Stuff | IoU | mIoU |
|---|---|---|---|---|---|---|---|---|---|---|---|---|
| 20 % | **15.38** | **12.74** | 32.76 | **30.85** | **4.31** | 32.80 | **9.92** | **18.88** | 32.74 | **46.08** | **40.90** | **15.33** |
| 50 % | 14.45 | 6.30 | **41.95** | 9.75 | 2.09 | **42.67** | 3.33 | 9.35 | **41.43** | 14.43 | **40.90** | **15.33** |

20 % IoU threshold for matching of ground-truth and predicted instances, to present representative evaluation results. However, in Tab. 9, we provide evaluation results of IPFormer under PanoSSC's relaxed 20 % IoU threshold. Expectedly, RQ metrics increase substantially, since more instances are recognized, while these are segmented with less fidelity. Thus, SQ metrics decrease.

## A.7  Additional Ablation Experiments

Tab. 10 presents extensive additional ablation experiments on the two-stage training, the dual-head architecture, the interplay between SSC and PSC, and sensitivity to hyperparameters. All findings are in line and underscore our findings discussed in Sec. 4.

**Two-Stage Training and Dual-Head Architecture.**    Single-head methods (p, q) struggle to balance SSC and PSC, with single-stage configuration (p) performing the worst due to its inability to separate semantic and instance-level learning. Single-stage, dual-head methods (a, b) also fall short, as the lack of stage-wise optimization hinders instance registration. Two-stage methods highlight the advantages of stage-wise training: SSC-focused approaches (e, f, k) excel in SSC but underperform in PSC due to limited adaptation, while PSC-prioritized methods (j, n) improve instance registration but compromise semantic consistency or balance. IPFormer, method (r), decouples SSC and PSC optimization, achieving strong instance registration and balanced performance across both tasks, with a moderate SSC trade-off.

**Interplay between SSC and PSC.**    Across the design space, methods that emphasize SSC (e, f, k) achieve strong semantic scores but degrade PSC performance, especially PQ-Thing. Conversely, approaches prioritizing PSC (j, n, q) boost PQ-Thing or PQ-Stuff at the cost of SSC quality or overall balance. Joint or single-head variants (a, b, p, q) further struggle with instance registration or overall consistency. In contrast, our dual-head, two-stage method (r) yields the best PQ-All and strong performance across PQ-Thing and PQ-Stuff, with only a minor SSC trade-off.

**Sensitivity to Hyperparameters.**    We train the SSC task in Stage 1 (Eq. 8) using established hyperparameters for the cross-entropy, Semantic-SCAL, and Geometric-SCAL cost functions (Sec. 3.6). All weights associated with these functions are set to 1, consistent with state-of-the-art and established SSC works, such as CGFormer [59], OccFormer [63], and MonoScene [4]. Nevertheless, we investigate the effect of removing the Depth loss, as its impact has not been extensively studied. Furthermore, we analyze the effect of varying hyperparameters of the Sigmoid Focal Loss [33], specifically designed to down-weight easier examples and focus the training process on harder-to-classify examples:

$$\mathrm{FL}(p) = -\alpha_t(1 - p_t)^\gamma \log(p_t), \tag{12}$$

where $p_t$ is the predicted probability for the true class after applying the sigmoid function, $\alpha_t$ is the class-balancing weight, and $\gamma$ controls the down-weighting of well-classified examples.

While methods (e) and (f) in Tab. 10 achieve the highest SSC performance in terms of mIoU and IoU, respectively, their PSC performance deteriorates significantly. A similar trend is observed for method (k), which attains the second-best results for both mIoU and IoU. In contrast, the best PSC performance, measured by PQ-Thing, is achieved by method (j). Although this method demonstrates satisfactory SSC performance, it suffers from reduced PQ-Stuff and, consequently, lower PQ-All performance. Moreover, the highest PQ$^\dagger$ and PQ-Stuff performance is achieved by methods (n) and (q), respectively. However, both methods exhibit a notable decline in PQ-Thing. Specifically, method (n) experiences a substantial reduction in SSC performance, whereas method (q) maintains adequate SSC scores.

Table 10: Further ablation experiments analyzing the sensitivity of SSC and PSC performance with respect to the hyperparameters of the objective functions, the dual-head architecture, and the two-stage training strategy. For hyperparameters, blue values indicate a difference from our proposed IPFormer configuration (r). For SSC and PSC metrics, **bold** and underlined values represent best and second-best results, respectively.

| | Head(s) | Stage 1 SSC Losses | | | | Stage 2 SSC Losses | | | Stage 2 PSC Losses | | | | | | SSC Metrics | | PSC Metrics | | | |
|---|---|---|---|---|---|---|---|---|---|---|---|---|---|---|---|---|---|---|---|---|
| | | Depth | CE | Sem | Geo | CE | Sem | Geo | Depth | CE | DICE | Focal | | | IoU | mIoU | PQ† | PQ-All | PQ-Thing | PQ-Stuff |
| | | $\lambda_{depth}$ | $\lambda_{ce}$ | $\lambda_{scal}^{sem}$ | $\lambda_{scal}^{geo}$ | $\lambda_{ce}$ | $\lambda_{scal}^{sem}$ | $\lambda_{scal}^{geo}$ | $\lambda_{depth}$ | $\lambda_{ce}$ | $\lambda_{dice}$ | $\lambda_{focal}$ | $\alpha$ | $\gamma$ | | | | | | |
| (a) | Dual | — | — | — | — | 1.00 | 1.00 | 1.00 | 0.0001 | 1.00 | 1.00 | 40.00 | 0.25 | 2.00 | 41.30 | 15.26 | 14.21 | 5.57 | 1.06 | 8.84 |
| (b) | Dual | — | — | — | — | 1.00 | 1.00 | 1.00 | 0.0001 | 1.00 | 1.00 | 50.00 | 0.21 | 2.30 | 39.45 | 13.06 | 14.45 | 5.03 | 0.99 | 7.79 |
| (c) | Dual | 0.0001 | 1.00 | 1.00 | 1.00 | 1.00 | 1.00 | 1.00 | 0.0001 | 1.00 | 1.00 | 40.00 | 0.25 | 2.00 | 43.44 | 15.79 | 14.35 | 6.06 | 1.66 | 9.27 |
| (d) | Dual | 0.0001 | 1.00 | 1.00 | 1.00 | 0.50 | 0.50 | 0.50 | 0.0001 | 1.00 | 1.00 | 40.00 | 0.25 | 2.00 | 43.62 | 15.30 | 14.42 | 6.27 | 1.88 | 9.46 |
| (e) | Dual | 0.0001 | 1.00 | 1.00 | 1.00 | 0.20 | 0.20 | 0.20 | 0.0001 | 1.00 | 1.00 | 40.00 | 0.25 | 2.00 | 43.99 | **16.29** | 14.46 | 6.05 | 1.60 | 9.29 |
| (f) | Dual | 0.0001 | 1.00 | 1.00 | 1.00 | 0.10 | 0.10 | 0.10 | 0.0001 | 1.00 | 1.00 | 40.00 | 0.25 | 2.00 | **44.13** | 16.11 | 14.41 | 6.08 | 1.72 | 9.26 |
| (g) | Dual | 0.0001 | 1.00 | 1.00 | 1.00 | 0.01 | 0.01 | 0.01 | 0.0001 | 1.00 | 1.00 | 40.00 | 0.25 | 2.00 | 43.94 | 15.04 | 14.67 | 5.16 | 0.90 | 8.27 |
| (h) | Dual | 0.0001 | 1.00 | 1.00 | 1.00 | 1.00 | 1.00 | 1.00 | 0.0001 | 1.00 | 0.00 | 40.00 | 0.25 | 2.00 | 43.34 | 15.83 | 13.89 | 5.33 | 0.49 | 8.86 |
| (i) | Dual | 0.0001 | 1.00 | 1.00 | 1.00 | 0.10 | 0.10 | 0.10 | 0.0001 | 1.00 | 0.10 | 40.00 | 0.25 | 2.00 | 42.97 | 15.64 | 14.11 | 5.59 | 1.27 | 8.74 |
| (j) | Dual | 0.0001 | 1.00 | 1.00 | 1.00 | 0.03 | 0.03 | 0.03 | 0.0001 | 1.00 | 1.00 | 50.00 | 0.21 | 2.30 | 43.53 | 15.65 | 14.39 | 6.19 | **2.38** | 8.96 |
| (k) | Dual | 0.0001 | 1.00 | 1.00 | 1.00 | 0.50 | 0.50 | 0.50 | 0.0001 | 1.00 | 1.00 | 50.00 | 0.21 | 2.30 | 44.07 | 16.25 | 14.23 | 5.97 | 1.67 | 9.11 |
| (l) | Dual | 0.0001 | 1.00 | 1.00 | 1.00 | — | — | — | 0.0001 | 1.00 | 2.00 | 40.00 | 0.25 | 2.00 | 32.45 | 7.28 | 13.92 | 5.92 | 1.58 | 9.08 |
| (m) | Dual | 0.0001 | 1.00 | 1.00 | 1.00 | — | — | — | 0.0001 | 1.00 | 1.00 | 50.00 | 0.21 | 2.30 | 40.45 | 14.74 | 14.59 | 5.74 | 1.53 | 8.80 |
| (n) | Dual | 0.0001 | 1.00 | 1.00 | 1.00 | — | — | — | 0.0001 | 1.00 | 1.00 | 45.00 | 0.21 | 2.30 | 36.73 | 11.69 | **15.14** | 4.43 | 0.78 | 7.08 |
| (o) | Dual | — | 1.00 | 1.00 | 1.00 | — | — | — | — | 1.00 | 1.00 | 40.00 | 0.25 | 2.00 | 41.16 | 14.90 | 14.42 | 6.15 | 1.68 | 9.40 |
| (p) | Single | 0.0001 | 1.00 | 1.00 | 1.00 | — | — | — | 0.0001 | 1.00 | 1.00 | 40.00 | 0.25 | 2.00 | 4.90 | 2.53 | 10.66 | 0.42 | 0.13 | 0.63 |
| (q) | Single | — | — | — | — | — | — | — | 0.0001 | 1.00 | 1.00 | 40.00 | 0.25 | 2.00 | 42.31 | 14.80 | 14.64 | 6.23 | 1.75 | **9.49** |
| (r) | Dual | 0.0001 | 1.00 | 1.00 | 1.00 | — | — | — | 0.0001 | 1.00 | 1.00 | 40.00 | 0.25 | 2.00 | 40.90 | 15.33 | 14.45 | **6.30** | 2.09 | 9.35 |

Finally, our proposed IPFormer configuration, method (r), achieves a balanced PSC performance by obtaining the best score for PQ-All and the second-best results for both PQ-Thing and PQ-Stuff, with a moderate trade-off in SSC performance.

