# OpenReview forum: "IPFormer: Visual 3D Panoptic Scene Completion with Context-Adaptive Instance Proposals"
_NeurIPS.cc/2025/Conference — NeurIPS 2025 poster_

### Official Review · Reviewer_Kgxr · 2025-07-01

**Clarity:** 2
**Significance:** 2
**Originality:** 3
**Rating:** 4
**Confidence:** 5

**Summary:**

This paper derives the instance queries from proposal generator with semantic scene completion. With known initialization, the proposed methods out perform without any clustering.

**Questions:**

1. What is the main contribution to reduce the runtime? No additional DBSCAN computation?
2. Should we consider the invisible voxel features? This might be occurs another inefficient computation and might mislead at instance-voxel alignment.  What about just drop the invisible voxel features? This might increase the efficiency of whole pipeline.
3. Could this also comparable in the Occ3d nuScene dataset? Especially for the SparseOcc that I've method the weakness.

**Ethical Concerns:**

["NO or VERY MINOR ethics concerns only"]

**Final Justification:**

As authors resolve my concerns and justify motivation especially for the instance proposals, I incline the rate as borderline accept.

**Limitations:**

Thanks for the contribution in autonomous vision community.
As the SemanticKITTI dataset is more difficult setup to improve the task. Author should formulate the problem with this hard condition(such as sparse, single view) or tackling the weaknesses like inefficiency pipeline (compare to the Mask2Former based architecture). But still I concern the problem itself is from the SemanticKITTI dataset not the architecture. So I suggest to demonstrate on the other dataset, and prove the problem is also valid.

**Paper Formatting Concerns:**

No paper formatting issue

**Quality:**

3

**Strengths And Weaknesses:**

Strengths : Better reconstruction quality for the thing-category instances.

Weaknesses :
1. Inefficient pipeline - The proposed network are working with two stages, one is for learning semantic scene completion to initialize the  instances, and with this IPFormer compute instance segmentation.

2. The problem is from the SemanticKITTI dataset not from the method.  As the ground truth gives misclassification and geometric ambiguity, instance queries might not aligned well as shown in Figure 3. If the method itself is matter, then the author should demonstrate that instance are not distinguished in the other clean dataset( such as Occ3D or nuCraft)

3. Not the first time for the Image only Panoptic Scene Completion.
SparseOcc* already proposed the Panoptic Scene Completion in Occ3D based on the nuScene dataset.
SparseOcc* : Fully Sparse 3D Panoptic Occupancy Prediction, ECCV'24

---

> ### Author Rebuttal · Authors · 2025-07-31
>
> > `W1:` Inefficient pipeline: Two-stage process for SSC and PSC.
>
> We appreciate your insightful feedback. To address your concerns, we conduct additional ablation experiments, shown in Tab. 1. We apologize for the limited formatting due to Markdown.
>
> Single-stage methods (p, q) struggle to balance SSC and PSC, with single-head configuration (p) performing the worst due to its inability to separate semantic and instance-level learning. Single-stage, dual-head methods (a, b) also fall short, as the lack of stage-wise optimization hinders instance registration. Two-stage methods highlight the advantages of stage-wise training: SSC-focused approaches (e, f, k) excel in SSC but underperform in PSC due to limited adaptation, while PSC-prioritized methods (j, n) improve instance registration but compromise semantic consistency or balance. IPFormer, method (r), decouples SSC and PSC optimization, achieving strong instance registration and balanced performance across both tasks, with a moderate SSC trade-off.
>
> We also improve IPFormer’s efficiency by integrating non-learnable operations, such as panoptic aggregation, noted as efficient by Reviewer [rTop]. Additionally, we use a single intermediate grid resolution, unlike methods like OccFormer and PaSCo, which rely on multi-scale voxel features.
>
> Table 1: Additional ablation experiments. Note that $\lambda_{\text{depth}}$ was mistakenly reported as $1$ in our manuscript, but is actually $0.0001$. We apologize.
> |     |Head(s)| 1st Stage |     |     |     | 2nd Stage |      |      |            |     |      |       |      |     |**Results**|       |               |        |          |          |
> |:----|:----- |:-----------|:----|:----|:----|:-----------|:-----|:-----|:-----------|:----|:-----|:------|:-----|:----|:--------------|:------|:--------------|:-------|:---------|:---------|
> |     |       | SSC Losses|     |     |     | SSC Losses |      |      | PSC Losses |     |      |       |      |     |SSC Metrics    |       |PSC Metrics    |        |          |
> |     |       | Depth      | CE  | Sem | Geo | CE         | Sem  | Geo  | Depth      | CE  | DICE | Focal |      |     | IoU           | mIoU  | PQ$^\dagger$| PQ-All | PQ-Thing | PQ-Stuff |
> |     |       | $λ$        | $λ$ | $λ$ | $λ$ | $λ$        | $λ$  | $λ$  | $λ$        | $λ$ | $λ$  | $λ$   | $α$  | $γ$ |               |       |               |        |          |          |
> | (a) | Dual  | **-**      |**-**|**-**|**-**| 1.00       | 1.00     | 1.00     | 0.0001     | 1   | 1    | 40.00   | 0.25   | 2.00   | 41.3          | 15.26 | 14.21         | 5.57   | 1.06     | 8.84     |
> | (b) | Dual  | **-**      |**-**|**-**|**-**| 1.00       |1.00      |1.00      | 0.0001 | 1   | 1    | **50.00**| **0.21**| **2.30** | 39.45         | 13.06 | 14.45         | 5.03   | 0.99     | 7.97     |
> | (c) | Dual  | 0.0001     | 1   | 1   | 1   | **1.00**   | **1.00** | **1.00** | 0.0001     | 1   | 1    | 40.00   | 0.25   | 2.00   | 43.44         | 15.79 | 14.35         | 6.06   | 1.66     | 9.27     |
> | (d) | Dual  | 0.0001     | 1   | 1   | 1   | **0.50**   | **0.50** | **0.50** | 0.0001     | 1   | 1    | 40.00   | 0.25   | 2.00   | 43.62         | 15.3  | 14.42         | **6.27** | 1.88     | _**9.46**_ |
> | (e) | Dual  | 0.0001     | 1   | 1   | 1   | **0.20**   | **0.20** | **0.20** | 0.0001     | 1   | 1    | 40.00   | 0.25   | 2.00   | 43.99         | **16.29** | 14.46         | 6.05   | 1.6      | 9.29     |
> | (f) | Dual  | 0.0001     | 1   | 1   | 1   | **0.10**   | **0.10** | **0.10** | 0.0001     | 1   | 1    | 40.00   | 0.25   | 2.00   | **44.13**     | 16.11 | 14.41         | 6.08   | 1.72     | 9.26     |
> | (g) | Dual  | 0.0001     | 1   | 1   | 1   | **0.01**   | **0.01** | **0.01** | 0.0001     | 1   | 1    | 40.00   | 0.25   | 2.00   | 43.94         | 15.04 | _**14.67**_     | 5.16   | 0.9      | 8.27     |
> | (h) | Dual  | 0.0001     | 1   | 1   | 1   | **1.00**   | **1.00** | **1.00** | 0.0001     | 1   | **0.00** | 40.00   | 0.25   | 2.00   | 43.34         | 15.83 | 13.89         | 5.33   | 0.49     | 8.86     |
> | (i) | Dual  | 0.0001     | 1   | 1   | 1   | **0.10**   | **0.10** | **0.10** | 0.0001     | 1   | **0.10** | 40.00   | 0.25   | 2.00   | 42.97         | 15.64 | 14.11         | 5.59   | 1.27     | 8.74     |
> | (j) | Dual  | 0.0001     | 1   | 1   | 1   | **0.03**   | **0.03** | **0.03** | 0.0001     | 1   | 1    | **50.00** | **0.21** | **2.30** | 43.53         | 15.65 | 14.39         | 6.19   | **2.38** | 8.96     |
> | (k) | Dual  | 0.0001     | 1   | 1   | 1   | **0.50**   | **0.50** | **0.50** | 0.0001     | 1   | 1    | **50.00** | **0.21** | **2.30** | _**44.07**_     | _**16.25**_ | 14.23         | 5.97   | 1.67     | 9.11     |
> | (l) | Dual  | 0.0001     | 1   | 1   | 1   | -          | -     | -     | 0.0001     | 1   | **2.00** | 40.00   | 0.25   | 2.00   | 32.45         | 7.28  | 13.92         | 5.92   | 1.58     | 9.08     |
> | (m) | Dual  | 0.0001     | 1   | 1   | 1   | -          | -     | -     | 0.0001     | 1   | 1    | **50.00** | **0.21** | **2.30** | 40.45         | 14.74 | 14.59         | 5.74   | 1.53     | 8.8      |
> | (n) | Dual  | 0.0001     | 1   | 1   | 1   | -          | -     | -     | 0.0001     | 1   | 1    | **45.00** | **0.21** | **2.30** | 36.73         | 11.69 | **15.14**     | 4.43   | 0.78     | 7.08     |
> | (o) | Dual  | **0.0000** | 1   | 1   | 1   | -          | -     | -     | **0.0000** | 1   | 1    | 40.00   | 0.25   | 2.00   |      41.16 | 14.90  |   14.41    |  6.15   |   1.68      |  9.40      |
> | (p) |**Single** |0.0001     | 1   | 1   | 1   | -          | -     | -     | 0.0001     | 1   | 1    | 40.00   | 0.25   | 2.00   | 4.90          | 2.53 | 10.66         | 0.42 | 0.13 | 0.63 |
> | (q) |**Single** | **-**      |**-**|**-**|**-**| -       | -     | -     | 0.0001     | 1   | 1    | 40.00   | 0.25   | 2.00   | 42.31          | 14.80 | 14.64         | 6.23   | 1.75     | **9.49**     |
> | (r) | Dual | 0.0001     | 1   | 1   | 1   | -          | -     | -     | 0.0001     | 1   | 1    | 40.00   | 0.25   | 2.00   | 40.9          | 15.33 | 14.45         | **6.30** | _**2.09**_ | 9.35 |
>
> > `W2:` Ground truth artifacts in SemanticKITTI and the need for evaluation on cleaner datasets.
>
> We thank you for noting that SemanticKITTI contains trace artifacts from aggregated LiDAR scans. To address this, we cross-validate IPFormer on the SSCBench-KITTI-360 dataset, a clean dataset without trace artifacts, similar to Occ3D and nuCraft. As shown in Tab. 1 of the rebuttal to Reviewer [rTop], IPFormer consistently outperforms its closest competitor, CGFormer+DBScan, across all PSC and SSC metrics, demonstrating that instance delineation in our method is not influenced by ground-truth artifacts.
>
> > `W3 & Q3:` Comparison with SparseOcc for image-only PSC on the Occ3D-nuScenes dataset.
>
> We appreciate your effort in highlighting the camera-based occupancy method SparseOcc. However, we argue that a comparison with IPFormer is not equitable. SparseOcc uses (i) $6$ surround-view cameras ($6$ multi-view images) and (ii) up to $96$ images per forward pass (8 or 16 frames, each with 6 images) to reconstruct a panoptic grid. Additionally, SparseOcc evaluates performance using non-standard metrics, differing from classical panoptic metrics established in the field.
>
> Regarding the Occ3D-nuScenes dataset, we kindly refer to Tab. 1 of the rebuttal to Reviewer [rTop], where we cross-validate IPFormer on the SSCBench-KITTI-360 dataset, a clean dataset without trace artifacts. These results on zero-shot generalization demonstrate that IPFormer consistently outperforms its closest competitor, CGFormer+DBScan, across all PSC and SSC metrics.
>
> > `Q1:` What is the main contribution to reduce the runtime? No additional DBSCAN computation?
>
> We thank you for raising these important questions. The main contribution of our work is not the removal of DBSCAN to reduce runtime, as this advantage is a byproduct of addressing PSC in an end-to-end fashion. Instead, our primary contribution lies in the introduction of context-adaptive instance proposals, which extend state-of-the-art instance queries that are static at test time.
>
> As shown in Tab. 2 of our manuscript, this approach improves Thing-related metrics by an average of $18.65$%. This highlights the effectiveness of context-adaptive proposals and marks a key advancement in 3D vision research.
>
> > `Q2:` Consideration of invisible voxel features and their impact on efficiency and instance-voxel alignment.
>
> We appreciate these insightful questions and provide additional ablation results in Tab. 2.
>
> Including invisible voxels demonstrates superior performance across all PSC metrics. For SSC metrics, the configuration without invisible voxels achieves slightly better IoU, while our method attains marginally higher mIoU. Importantly, GPU utilization remains unchanged, as IPFormer operates on a dense voxel grid.
>
> Table 2: Ablation experiments on the influence of invisible voxels, on the SemanticKITTI dataset.
> | Method          | PQ$^\dagger$ | PQ-All | SQ-All       | RQ-All | PQ-Thing | SQ-Thing     | RQ-Thing | PQ-Stuff | SQ-Stuff     | RQ-Stuff | IoU   | mIoU        |
> |-----------------|-------------|--------|--------------|--------|----------|--------------|----------|----------|--------------|----------|-------|-------------|
> | w/o invisible voxels | 14.28       | 6.00   | 33.20        | 9.33   | 1.69     | 28.77        | 2.66     | 9.13     | 36.42        | 14.18    | **41.38**  | 15.05        |
> | w/ invisible voxels  | **14.45**       | **6.30** | **41.95**   | **9.75** | **2.09** | **42.67**   | **3.33** | **9.35** | **41.43**   | **14.43** | 40.90 | **15.33** |
>
> > Closing Remark.
>
> The Reviewer's effort and constructive suggestions are sincerely appreciated. We hope our efforts are acknowledged by the Reviewer and encourage an upgrade of the rating. Should there be any remaining questions, we would be happy to provide further clarification.

---

> > ### Comment · Reviewer_Kgxr · 2025-08-05
> >
> > Thank you for your clarification in the rebuttal. While some of my concerns have been addressed, a few still remain.
> >
> > I am still not fully convinced of the novelty of the proposed method compared to other panoptic approaches such as SparseOcc and PaSCo (excluding the LiDAR setup), as also pointed out by Reviewer 8FSR. Although SparseOcc uses a different input configuration, its core methodology and motivation appear similar.
> >
> > Even if the proposed method operates on dense voxels, I remain unconvinced that abandoning the advantages of sparse voxel representations is justified. It is unclear whether working with dense voxels provides sufficient benefits to outweigh the efficiency and scalability advantages of sparse representations. Perhaps the authors should instead focus on addressing the challenges inherent to single-image setups.
> >
> > Given these concerns, I remain hesitant to increase my score at this time.

---

> ### Author Response · Authors · 2025-08-05
> **Response to Reviewer Kgxr**
>
> Thank you for your comment. We are glad that some of the concerns could be addressed, and are pleased to provide clarification on the remaining concerns.
>
> **Novelty of our method compared to SparseOcc and PaSCo**:
>
> Unlike our method, the core methodological contributions of SparseOcc and PaSCo (apart from PaSCo being LiDAR-based), primarily address computational efficiency by leveraging sparsity mechanisms. PaSCo employs a multi-scale sparse generative decoder to reconstruct scene geometry and semantics while relying on static queries (called “query features” in the underlying Mask2Former decoder architecture) for instance prediction. Similarly, SparseOcc introduces a sparse voxel decoder and sparse instance queries to predict semantic and instance occupancy. However, a key limitation of both methods is their reliance on static queries, which are fixed during inference and cannot adapt to the observed scene. IPFormer directly addresses this limitation by introducing context-adaptive instance proposals that dynamically adapt to the scene at both training and test time. This dynamic adaptation allows IPFormer to better capture scene-specific details and improve instance-level reasoning. Therefore, the motivation of our work is different, as we aim to overcome the rigidity of static queries. Our core methodological contribution lies in introducing context-adaptive instance proposals, which alleviate the limitations posed by static queries in both PaSCo and SparseOcc, marking a significant advancement in 3D vision research.
>
> To this end, Reviewer [8FSR] does not question the novelty of our approach but actually highlights it as a strength. Specifically, Reviewer [8FSR] states that
>
> > DETR-style queries, which are learned during training and fixed at test time, have limited representational capacity and ignore input image's context which may hurt generalization. Therefore the proposed dynamically adaptive instance proposals which are generated from the input image context seems like a sound improvement over DETR-style instance queries. This improvement is backed by the ablation in Table 2 which shows consistent improvement in almost all metrics when switching from learned but fixed instance queries to their proposed context adaptive instance proposals.”
>
> This feedback, along with Reviewer [rTOP]’s acknowledgment of “addressing limitations of static transformer queries”, underscores the novelty and methodological contribution of our work.
>
> **Sparse voxel representations:**
>
> As discussed above, our primary goal is to address the limitations of static instance queries by introducing context-adaptive instance proposals for vision-based PSC. The use of dense voxels is a deliberate design choice to ensure effective reasoning about voxel-instance relationships, which is successfully demonstrated by the ablation experiment presented in Tab. 2 of the rebuttal. Incorporating sparsity mechanisms, while beneficial in other contexts such as SparseOcc’s handling of up to 96 temporal frames, would diverge from our primary research objective of evaluating the effectiveness of context-adaptive instance proposals. By isolating our investigations to this specific contribution, we ensure scientific rigor, allowing us to clearly demonstrate the advantages of our proposed solution. This focused approach ensures that our contributions are both targeted and impactful, as evidenced by our extensive ablation studies, the state-of-the-art performance on panoptic metrics for in-domain data, superior performance on out-of-domain data via zero-shot generalization, and the runtime reduction of over 14× compared to baseline methods, underscoring both the impact and efficiency of our solution.
>
> ---
>
> We hope that the remaining concerns are clarified, and we will incorporate these clarifications into the revised manuscript, as your feedback is valuable to improve the quality of our work. Please let us know if any concerns remain, as we would be pleased to provide further clarification.

---

> > ### Comment · Reviewer_Kgxr · 2025-08-06
> >
> > Thank you for your detailed clarification. After revisiting the papers, especially the part regarding proposal initialization, I am inclined to raise my score to a weak accept. It would be helpful to include a qualitative comparison of query activations between static instance queries and the proposed instance proposals.

---

> > > ### Author Response · Authors · 2025-08-06
> > > **Official Comment by Authors**
> > >
> > > Dear Reviewer Kgxr,
> > >
> > > We are glad that we were able to address your remaining concerns and that you are inclined to raise your score. We greatly appreciate the time and effort you dedicated to revisiting the papers.
> > >
> > > Additionally, we are pleased to include a qualitative comparison of query activations between static instance queries and the proposed instance proposals.

---

### Official Review · Reviewer_8FSR · 2025-07-02

**Clarity:** 2
**Significance:** 2
**Originality:** 2
**Rating:** 4
**Confidence:** 4

**Summary:**

The authors propose IPFormer, a monocular image-based 3D panoptic scene completion method which jointly recovers the complete 3D geometry of a scene while identifying individual objects and their semantics from a single RGB image. To improve upon previous works which learn query proposals during training and fix them at test time, they propose to generate instance proposals from an input image's context at both training and testing, enabling better identification and localization of object instances. Furthermore, a two stage training pipeline is proposed where the semantic scene completion task is first optimized for to learn semantics and complete geometry and then the panoptic scene completion task is optimized to identify individual object instances.

**Questions:**

**Questions:**
- The definition of $S$ in line 151 is confusing to me. Why is the quantization of the 3D feature $F_{3D}$ a point coordinate $(x, y, z)$? The point coordinate should only be dependent on $(u, v, d)$ and has nothing to do with the feature $F_{3D}$.

**Suggestions:**
- In Figure 2, it would be better to have an additional arrow going from the RGB image to depth map to show that the depth is estimated. The current figure makes it seem as if the method is an RGBD method instead of a RGB method.
- The description of 3D deformable cross attention in Eq. 3 and lines 172-175 could be improved. There is no mention of what *reference* points and *sampled* points are. Furthermore, $\Pi$ is defined as the projection from 3D to 2D in line 160 but then seems to be redefined to mean something else in line 172.
- In line 265, I think it was meant to say “Stuff-instances” not “Thing-instances”

**Ethical Concerns:**

["NO or VERY MINOR ethics concerns only"]

**Final Justification:**

After looking over the rebuttals and further discussion with the authors, most of the concerns I had have been addressed by the authors. Therefore, I have chosen to raise my score to borderline accept. While the work has limited novelty, the proposed method boasts a large speed up improvement alongside a moderate performance improvement over existing solutions to camera-based panoptic scene completion, making it overall an interesting work.

**Limitations:**

yes

**Quality:**

2

**Strengths And Weaknesses:**

**Strengths:**
- DETR-style queries, which are learned during training and fixed at test time, have limited representational capacity and ignore input image's context which may hurt generalization. Therefore the proposed dynamically adaptive instance proposals which are generated from the input image context seems like a sound improvement over DETR-style instance queries. This improvement is backed by the ablation in Table 2 which shows consistent improvement in almost all metrics when switching from learned but fixed instance queries to their proposed context adaptive instance proposals.
- As the proposed method directly predicts instance segmentations, it offers a significant speed up (14x) over the monocular semantic scene completion works which require running DBScan for identifying object instances.

**Weaknesses:**
- Instance proposal generation is the main contribution of this work yet there is no description of what $g$ is in Eq. 5. It mentions inspiration from CGFormer's query generator, but from my understanding CGFormer’s query generator generates a dense voxel grid of queries and then does a depth-based query proposal. When comparing architectures to CGFormer, these two steps are $V$ from Eq. 2 and $\tilde{V_{vis}}$ from Eq. 3 in this work, so it's not clear how CGFormer's query generator is adapted to generate $I_{p}$. The technical details of the contribution needed to be more clearly stated.
- Despite being trained for the panoptic scene completion task, IPFormer barely outperforms CGFormer + DBScan which is not trained for panoptic scene completion. Looking at Table 1 in the main paper and Table 6 in the appendix, this boost in performance really only comes from better RQ in *stuff* categories (e.g., sidewalks, vegetation). Alternatively, CGFormer actually outperforms the proposed IPFormer on segmenting individual object instances of the *thing* categories. Moreover, this small boost in performance comes at a cost of a large drop in performance on SSC metrics as shown in Table 8 of the appendix.
- Since the work is heavily based off of other works (CGFormer, PaSCo), the methodology section could be written a little more clearly to identify what is from those works and what is unique or a contribution from the proposed approach. For example, the "voxel visibility" and "voxel proposals" from Section 3.2 are essentially straight from CGFormer but there is no mention of this. Section 3.4 could use a better transition between what part of the encoder is from CGFormer and what is not. For Section 3.5, it is unclear whether this panoptic aggregation is just directly from PaSCo or if there are differences.

---

> ### Author Rebuttal · Authors · 2025-07-31
>
> We are sincerely thankful for your valuable feedback and thoughtful suggestions, as well as for taking the time and effort to thoroughly understand the detailed technical intricacies of our work – we deeply appreciate it.
>
> > `W1:` Clarification on instance proposal generation and its adaptation from CGFormer.
>
> We thank you for pointing out the need for additional clarity regarding the technical details of our instance proposal generation. We acknowledge that the formulation in our manuscript could have been more explicit and are pleased to provide further elaboration.
>
> CGFormer’s query generator performs two key steps: (1) it generates a dense voxel grid of context-dependent queries, and (2) enhances their representational capacity by adding learnable embeddings. This understanding is derived from CGFormer’s official implementation on GitHub, as the details are not fully described in the paper. In our work, we extend these ideas to a novel domain by introducing the first context-adaptive instance proposal mechanism for PSC. Specifically, we enhance our proposal generator by integrating learnable embeddings to improve its representational capacity. This approach represents a significant advancement in 3D vision research, as no prior work has explored context-adaptive instance proposals for PSC. Technically, our proposal generator $g$ (Eq. 5 in the manuscript) can be more precisely expressed as:
>
> \begin{equation}
> \mathbf{I}_P = g(\text{DSA}(\tilde{\mathbf{V}}\_\text{vis}\^\mathbf{x},\tilde{\mathbf{V}}\_\text{vis},\mathbf{x})) = \text{DSA}(\tilde{\mathbf{V}}\_\text{vis}\^\mathbf{x},\tilde{\mathbf{V}}\_\text{vis},\mathbf{x}) + \mathbf{W}\_\mathbf{I}
> \end{equation}
>
> To validate the effectiveness of this approach, we conduct an additional ablation experiment to analyze the performance of context-adaptive instance proposals with and without learnable embeddings. These results, presented in Tab. 1, demonstrate that the inclusion of learnable embeddings consistently improves SSC and PSC metrics, further supporting the value of this design choice.
>
> Table 1: Ablation on IPFormer’s instance proposals wiht and without the addition of learnable embeddings, on the SemanticKITTI dataset.
> | Method          | PQ$^\dagger$ | PQ-All | SQ-All       | RQ-All | PQ-Thing | SQ-Thing     | RQ-Thing | PQ-Stuff | SQ-Stuff     | RQ-Stuff | IoU   | mIoU        |
> |-----------------|-------------|--------|--------------|--------|----------|--------------|----------|----------|--------------|----------|-------|-------------|
> | w/o embeddings | **14.53**       | 6.04   | 33.26        | 9.43   | 1.73     | 22.37        | 2.68     | 9.18     | 41.19        | 14.33    | 39.9  | 14.8        |
> | w/ embeddings  | 14.45       | **6.30** | **41.95**   | **9.75** | **2.09** | **42.67**   | **3.33** | **9.35** | **41.43**   | **14.43** | **40.90** | **15.33** |
>
> > `W2:` Performance trade-offs between IPFormer and CGFormer across PSC and SSC metrics.
>
> We appreciate the insightful comments on the performance comparison between IPFormer and CGFormer+DBScan. To address this concern, we want to emphasize the following points:
>
> CGFormer itself does not outperform IPFormer. Its performance gains stem from the addition of DBScan clustering, which enables instance registration within the SSC prediction. For end-to-end solutions like IPFormer, the difference in SSC and PSC performance is primarily influenced by their distinct objectives. This observation, where SSC performance may be suboptimal despite balanced PSC performance, aligns with findings reported in PaSCo (CVPR 2024), the method that introduced the PSC task. The authors of PaSCo note that SSC and PSC metrics are not directly correlated, a conclusion further supported and validated by our findings. Additionally, since PaSCo is LiDAR-based, our method provides the insight that this phenomenon appears to extend across distinct data modalities.
>
> A more nuanced interpretation of the performance comparison is that CGFormer dedicates its capacity to optimizing SSC, while DBScan’s exhaustive deterministic clustering facilitates instance registration. In contrast, IPFormer addresses PSC in an end-to-end fashion, by dedicating its capacity to both the SSC and PSC objectives. While this leads to performance results that appear less favorable for SSC, IPFormer surpasses CGFormer+DBScan in primary panoptic metrics PQ$^\dagger$ and PQ-All, performs on par in individual metrics, and achieves a significant runtime reduction of over 14$\times$.
>
> > `W3:` Clarification of contributions and distinctions from CGFormer and PaSCo in the methodology.
>
> We are thankful for the careful reading of our manuscript and acknowledge the opportunity to clarify the distinctions between our contributions and prior works.
>
> Regarding the voxel visibility and voxel proposals in Section 3.3, we agree that voxel visibility could be cited more precisely, as it is a well-established process in existing works on voxel-based scene understanding, including CGFormer, OccFormer, Symphonies, and VoxFormer. In contrast, voxel proposals are explicitly stated in line 191 of our manuscript to align with the works referenced therein.
>
> For Section 3.4, we note in line 197 that the 3D Local and Global Encoder is based on CGFormer. This is the only component of the IPFormer encoder based on prior works, which is why no additional references are mentioned in this section. In the context of Section 3.5, we indicate in line 224 that the panoptic aggregation is inspired by PaSCo. However, our implementation extends this approach to operate on dense voxels without incorporating uncertainty calculations, which represents a key difference from PaSCo.
>
> We are pleased to provide these clarifications and will incorporate the details into the revised version of our manuscript to improve clarity and attribution.
>
> > `Q1:` Clarification of the definition and role of $S$ in relation to $\mathbf{F}_{3D}$ and point coordinates.
>
> We appreaciate your effort to point out the confusion regarding the definition of $S$ in line 151 and are pleased to clarify this aspect of our methodology.
>
> We confirm that the point coordinates $(x, y, z)$ are determined by the pixel coordinates $(u, v)$ and the depth bin $d$, independent of the feature $\mathbf{F}_{\text{3D}}$. The correct notation is therefore given by $S = ${$(u,v,d) \mid \mathcal{Q}((u,v,d)) = (x,y,z)$}.
>
> We will revise the manuscript to ensure that the distinction between spatial quantization and feature aggregation is more explicit.
>
> > `S1:` Suggestion to update Fig. 2 to clarify depth estimation from RGB input.
>
> Thanks for this suggestion. We will update Fig. 2 in the revised manuscript to include an arrow from the RGB image to the depth map, clarifying that the depth is estimated, and ensuring to properly convey that our method is RGB-based, not RGBD-based.
>
> >`S2:` Improvement of the description of 3D deformable cross-attention and clarification of reference points, sampled points, and the definition of $\mathbf{\Pi}$.
>
> We thank you for pointing out this potential ambiguity and are glad to clarify.
>
> In deformable cross-attention, "reference points" are predefined query locations in the target space ($\mathbf{V}\_\text{vis}$), and "sampled points" are specific locations in the source space ($\mathbf{F}_{\text{3D}}$) around the reference points, selected based on learned offsets to aggregate relevant information.
>
> Furthermore, in our manuscript, $\Pi$ consistently refers to the projection function, mapping 3D query positions $\mathbf{x}$ to their corresponding 2D reference points. This projected reference point $\Pi(\mathbf{x})$ is a 2D location on the feature map $F_{\text{3D}}$, which corresponds to the $(u, v)$ coordinates in the image plane. The offsets $\Delta \mathbf{p}$ are continuous displacement vectors calculated relative to the projected reference point $\Pi(\mathbf{x})$, enabling the model to sample features flexibly from locations in the 3D feature map using trilinear interpolation, which is denoted as $\psi(\cdot)$ in Eq. 5 of the manuscript. We will revise the manuscript to clarify this and ensure consistency in the explanation of $\Pi$.
>
> > `S3:` In line 265, I think it was meant to say “Stuff-instances” not “Thing-instances”
>
> Thank you for the careful reading. We agree that the term should indeed be "Stuff-instances" rather than "Thing-instances". We will correct this in the revised manuscript and appreciate the Reviewer’s attention to detail.
>
> > Closing Statement.
>
> We deeply appreciate the Reviewer's thorough analysis and exceptional attention to technical detail, as well as the constructive suggestions, which we will thoughtfully incorporate into the revised version of our manuscript. In light of the new evidence regarding **(1) comprehensive clarifications** on our contributions, technical details, and performance trade-offs, **(2) extensive additional ablation experiments** (see Tab. 1 in the rebuttal to Reviewer [Kgxr]), and **(3) additional results on zero-shot generalization** (see Tab. 1 in the rebuttal to Reviewer [rTop]), we are confident that our work represents a significant advancement in 3D vision research.
>
> We hope these efforts are acknowledged by the Reviewer and encourage an upgrade of the rating. Should there be any remaining questions, we would be happy to provide further clarification.

---

> > ### Comment · Reviewer_8FSR · 2025-08-04
> > **Response to authors**
> >
> > Thank you for the response. My concerns about writing clarity/details have been addressed; however, the provided response has not cleared up my concerns in regard to performance.
> >
> > **Performance against CGFormer:**
> > I understand that CGFormer itself is not a PSC method and requires the additional usage of DBSCAN to recover instances, but that is exactly my concern. A method that was trained for a different task and was simply post-processed by a heuristic-based clustering algorithm can obtain the same performance on the PSC task as the proposed approach which is specifically trained for the PSC objective. The performance improvement of the proposed approach over CGFormer+DBSCAN on $PQ^{\dagger}$ is insignificant, while the performance improvement on $PQ$ is also small. Moreover, these improvements in $PQ$ are solely due to better performance on $RQ$-Stuff. CGFormer+DBSCAN outperforms the proposed approach on $SQ$-stuff and the instance related metrics $PQ$-thing, $SQ$-thing, $RQ$-thing. Moreover, CGFormer significantly outperforms the proposed approach in SSC metrics. Therefore, the only advantage the proposed method really has over CGFormer is inference speed since it does not require the additional clustering post-processing.
> >
> > **Tradeoff between PSC and SSC:**
> > SSC performance being suboptimal in order to balance PSC performance does not align with the findings of PaSCo. Taken directly from their paper, "Incidentally, we note that PSC and SSC metrics are not directly correlated since we improve the former drastically".  No where do they state SSC gets worse in order to improve PSC. On SemanticKITTI, they report improvements of 35% on $PQ^{\dagger}$, 51% on $PQ$, and 8% on $mIoU$ over the best performing lidar-based SSC+DBSCAN method. Alternatively, the proposed approach reports improvements of 0.4% on $PQ^{\dagger}$ and 2.2% on $PQ$ while suffering from a relative decrease of 9% on $mIoU$ over the best performing camera-based SSC method (CGFormer). These results tell two different stories. PaSCo results imply that end-to-end PSC training can result in large improvements on PSC performance but less improvement on SSC performance over SSC methods, while the proposed approach suggests that end-to-end PSC training offers little to no improvement over SSC methods post-processed with DBSCAN, while suffering from performance drops in SSC.

---

> ### Author Response · Authors · 2025-08-05
> **Response to Reviewer 8FSR**
>
> Thank you for your response. We are pleased that most of the concerns have been addressed and appreciate the opportunity to clarify the remaining questions regarding performance.
>
> **Performance against CGFormer:**
>
> Before addressing the concerns, it is important to highlight that DBScan clustering is the currently established method for generating ground truth (GT) for PSC from GT SSC. Given the absence of established baselines for vision-based PSC, we adopt the approach proposed in LiDAR-based PaSCo, in which vision-based SSC is combined with DBScan to create reliable proxies. However, it is crucial to note that these proxies hold no practical relevance, due to their significant computational inefficiency.
>
> We confirm that both CGFormer+DBScan and IPFormer achieve comparable state-of-the-art performance on PSC metrics, with IPFormer exhibiting suboptimal results on SSC metrics. However, labeling CGFormer as being trained for a "different task" is not entirely accurate. PSC is inherently a generalization of SSC, where SSC tasks (occupancy + semantics) form a strict subset of PSC tasks (occupancy + semantics + instance registration). In baseline methods like CGFormer+DBScan, the PSC objective is achieved by first optimizing for the SSC task, followed by instance registration through DBScan clustering. Crucially, the PSC performance of CGFormer+DBScan is directly dependent on the quality of its SSC predictions, as DBScan itself is a heuristic-based clustering algorithm with no independent learning capacity. This dependency highlights that CGFormer+DBScan’s performance on PSC cannot be attributed solely to the clustering step but is fundamentally tied to the strength of its SSC outputs.
>
> To further contextualize this dependency, we emphasize the importance of zero-shot generalization. As presented in Tab. 1 of the rebuttal to Reviewer [rTop], our method significantly outperforms CGFormer+DBScan in both PSC and SSC metrics on out-of-domain data, with the exception of a marginal difference of 0.01% in $PQ$-Thing and $RQ$-Thing. Importantly, metrics initially reported as weaker on in-domain data, such as $SQ$-Thing and $SQ$-Stuff, demonstrate substantial improvements with our method when evaluated on out-of-domain data. Specifically, $SQ$-Thing improves from 20.06 for CGFormer+DBScan to 22.76 for our method, while $SQ$-Stuff improves from 16.19 to 25.89. This trend is also reflected in SSC metrics, where our approach outperforms CGFormer+DBScan on out-of-domain data, despite being outperformed on in-domain data. Moreover, analyzing the relative generalization gap, as also presented in Tab. 1, further underscores these findings. Across all PSC and SSC metrics, our method exhibits superior relative generalization, with the most substantial improvements observed in $SQ$-Thing, $SQ$-Stuff, $IoU$, and $mIoU$ metrics.
>
> These results highlight a critical limitation of the baselines, such as CGFormer+DBScan: their reliance on SSC predictions and heuristic clustering limit their ability to generalize to in-the-wild data. In contrast, our method not only improves efficiency by reducing inference time by over 14$\times$, but also achieves superior zero-shot generalization, surpassing CGFormer+DBScan in both absolute and relative PSC and SSC metrics on out-of-domain data. By providing a unified, end-to-end solution, our approach addresses the inherent limitations of heuristic-based PSC methods, delivering both competitive in-domain performance and superior generalization in real-world scenarios.
>
> **Tradeoff between PSC and SSC:**
>
> We acknowledge that our earlier phrasing may have unintentionally caused confusion regarding the findings in PaSCo, and we apologize for any misunderstandings. To clarify, while CGFormer+DBSCAN and our method exhibit comparable performance on PSC metrics, there is a notable discrepancy in SSC performance, with our method performing suboptimal. This outcome is unexpected, as similar PSC performance might intuitively suggest comparable SSC performance. However, this discrepancy generally aligns with the conclusion in PaSCo, which states that PSC and SSC metrics are not directly correlated. Furthermore, our zero-shot generalization results (Tab. 1 in the rebuttal to Reviewer [rTop]) qualitatively align with the findings in PaSCo. Specifically, while our method demonstrates significant improvements in PSC performance compared to CGFormer+DBSCAN, the gains in SSC performance remain marginal. These findings reinforce the notion that SC and SSC objectives are not inherently linked and further support the conclusions drawn in PaSCo.
>
> ---
>
> We will incorporate these clarifications into the revised manuscript, as your feedback is valuable to improve the quality of our work. Please let us know if you have any additional questions; we would be pleased to provide further clarification. Thank you for your time and effort.

---

> ### Comment · Reviewer_8FSR · 2025-08-07
> **Response to authors**
>
> Thank you for pointing out the additional table with further results. These findings on generalization ability certainly do help highlight the strengths of the proposed approach over CGFormer+DBSCAN. Please do include these results in the main paper, as without them, the statement that "PSC performance of CGFormer+DBSCAN is directly dependent on the quality of SSC predictions" works against the proposed method rather than for it (my major concern was exactly that a method trained for SSC and post-processed could perform as well as a method explicitly trained for PSC). As this has helped address my major concern, I have chosen to raise my score to borderline accept.
>
> With that being said, the stated relationship between PSC and SSC by the authors is still contradictory. An argument is still trying to be made that PSC and SSC performance are not directly correlated, yet it has been stated that "PSC is inherently a generalization of SSC" and the proposed PSC method is first trained with SSC objectives. Moreover, the results themself are mixed and inconclusive as to what is the exact relationship between PSC and SSC performance. I suggest dropping this conclusion/finding from the paper as it seems forced just as an attempt to align with a previous work, and also just leads to more confusion in regards to task and motivation (e.g., why are they not correlated in some manner if PSC is a generalization of SSC? why is the proposed method trained with SSC objectives if SSC isn't relevant to PSC performance?).

---

> > ### Author Response · Authors · 2025-08-07
> > **Response to Reviewer 8FSR**
> >
> > We are glad that we could alleviate your major concern, and are thankful that you have chosen to raise the score to borderline accept. Moreover, we will include the results on generalization ability in the main paper. Thank you for the time and effort you dedicate to this review.
> >
> > Regarding the relationship between PSC and SSC metrics, we will follow your suggestion and exclude the corresponding conclusion from our manuscript. However, your insightful comments and questions are inspiring and have encouraged us to continue this open discussion. We believe that the relationship between the two objectives depends on how the PSC metrics are computed. Specifically, $RQ$ is influenced by the IoU threshold $th$, which determines true positives (TPs), false positives (FPs), and false negatives (FNs) for a given set of predicted masks and ground-truth masks.
> >
> > Before providing a qualitative example in this context, recall the following: If a pair of predicted and ground-truth masks results in a TP (i.e., their $IoU > th$), then $RQ > 0$ and, consequently, $SQ > th$. This is evident in the class-wise PSC metrics in Tab. 6 of the technical appendix, where $th = 50$%. In other words, a lower $th$ increases $RQ$, as instances are more easily recognized, but their segmentation fidelity decreases, leading to a reduction in $SQ$ - and vice versa. This behavior is observed in Tab. 2 of the rebuttal to Reviewer [rTop], where we compare our method using the established 50% threshold with a relaxed 20% threshold.
> >
> > Now consider a scene consisting of only a single car, where the predicted mask is compared with the ground-truth mask of that car:
> >
> > 1. If the mask overlap is just above the IoU threshold, then $RQ > 0$, and thus $SQ > th$ (as discussed above).
> > 2. If the mask overlap is just below the IoU threshold, then $RQ = 0$ and $SQ = 0$.
> >
> > To this end, the PSC metrics of both cases can differ significantly, depending on whether the IoU threshold is exceeded. However, in both cases, the SSC metrics would remain very similar, as the overlaps between the masks are close to the threshold, meaning that the SSC metrics focus on semantic overlap rather than instance matching. Note that the above example considers a single instance of a single class. In practice, the PSC metrics of all instances of a class are aggregated, and the results are averaged across all classes to compute the final PSC metrics, which implies that the described effect can be amplified. In conclusion, the relationship between PSC and SSC metrics can be complex, depending on the IoU threshold and the margins relative to that threshold.

---

> > > ### Comment · Reviewer_8FSR · 2025-08-09
> > > **Response to authors**
> > >
> > > Thank you for this additional discussion and example. This certainly helps highlight the nuances between PSC and SSC metrics and why improved performance in PSC may not necessarily imply improved performance in SSC.

---

### Official Review · Reviewer_rTop · 2025-07-03

**Clarity:** 3
**Significance:** 3
**Originality:** 3
**Rating:** 4
**Confidence:** 3

**Summary:**

​This paper proposes IPFormer, a novel approach to ​​vision-based 3D Panoptic Scene Completion (PSC)​​ by leveraging ​​context-adaptive instance proposals​​ dynamically derived from image context. The contributions of this paper can be summarized as follows.

- Visibility-based sampling​​ (Sec. 3.3): Initializes instance proposals from visible voxels using camera depth and image features, ensuring dynamic adaptation to scene content at test time.

- Dual-head architecture​​ (Sec. 3.1, Table 5): Separates semantic scene completion (SSC) and PSC tasks via a two-stage training strategy, guiding latent space toward geometry/semantics before instance registration.

- Efficient PSC aggregation​​ (Sec. 3.5): Uses instance-voxel affinity scores to align semantics and instances without learnable parameters.

- State-of-the-art results​​: Outperforms SSC baselines + clustering (e.g., CGFormer + DBSCAN) in PQ (14.45) and PQ-All (6.30) on SemanticKITTI, with ​​14× faster inference​​ (0.33s vs. 4.51s).

**Questions:**

- How would IPFormer handle ​​outdoor scenes​​ with varying lighting/weather? Could visibility sampling fail for heavily occluded objects? How about the zero-shot generalization ability on in-the-wild data?

- Why use ​​dual-head training​​ instead of end-to-end joint optimization? Does stage-wise training limit cross-task synergy?

- No comparison with ​​PanoSSC​​ (vision-based PSC SOTA) due to inaccessible dataset (Sec 4.1). How would IPFormer fare under their relaxed IoU threshold? How's the progress now as the authors claim that this will be added to the final version of the paper?

- Can the approach scale to ​​higher resolutions​​ (e.g., 512^3 voxels) without prohibitive memory costs?

**Ethical Concerns:**

["NO or VERY MINOR ethics concerns only"]

**Final Justification:**

I keep my original rating.

**Limitations:**

yes

**Quality:**

3

**Strengths And Weaknesses:**

**Strengths**

- To my knowledge (which might be limited), this is the first method to use ​​test-time adaptive proposals​​ for vision-based PSC, addressing limitations of static transformer queries

- Adaptive proposals boost Thing-metrics by ​​18.65% avg​​ and PQ-All by ​​3.62%​​ versus random initialization

- Clear exposition, but the writing flow could be improved for easier understanding.

**Weaknesses**

- Struggles with rare classes (e.g., Bicyclist: 0.95% in Table 6) due to training data bias?
- High memory usage (52.8GB vs. 17.3GB for OccFormer, Table 7) limits accessibility. The proposed method is not that easy to train, advanced GPU such as A100/H100 is required.
- ​​Dependency on 2D models​​: Depth estimation (MobileStereoNet) and features (EfficientNet) may propagate errors to 3D. I am not sure if the depth estimation is overfit to KITTI datset, and wonder the performance on real-world data (zero-shot generalization?)

---

> ### Author Rebuttal · Authors · 2025-07-31
>
> We sincerely appreciate your insightful feedback and constructive suggestions. We address the concerns thoroughly.
>
> > ` W1`: Struggles with rare classes.
>
> We thank you for your detailed observation. As shown in Tab. 6 of our manuscript, this issue affects all methods and arises from the significant class imbalance in the SemanticKITTI dataset, where Thing classes make up only 4.53% of all voxels.
>
> We kindly direct you to Sec. `W2` of the rebuttal to Reviewer [z9JQ], where we elaborate on countermeasures.
>
> > `W2 & Q4:` High memory usage limits accessibility & scalability to higher voxel grid resolutions.
>
> We value your concern regarding the high memory usage and its implications on increased voxel grid resolution. We agree that this can be a limitation.
>
> System tools report $52.8$ GB memory usage due to CUDA overhead and preallocation, while detailed profiling shows the model itself uses $35$ GB. Unlike IPFormer, baselines like OccFormer do not perform PSC directly and require post-hoc clustering, adding extra computation. Thus, comparing GPU memory across SSC and PSC methods is not an entirely equitable assessment.
>
> Noteably, reliance on advanced hardware such as A100 GPUs has become increasingly common in recent 3D vision research. For instance, FB-OCC (CVPR 2023) [1] and VGGT (CVPR 2025) [2] were trained on 32 and 64 A100s, respectively. In comparison, IPFormer operates on a significantly lower computational budget, positioning it on the more efficient end of the spectrum while still achieving competitive performance.
>
> > `W3:` Dependency on 2D models, error propagation to 3D, and generalization beyond KITTI.
>
> We appreciate raising this concern. MobileStereoNet3D undergoes robust training to enhance generalizability. The model is pre-trained on the large synthetic SceneFlow dataset [3] (approx. 40,000 samples). Subsequently, the model is fine-tuned on two real-world datasets: DrivingStereo [4] (approx. 181,000 samples) and KITTI Scene Flow 2015 (KITTI2015) [5] (~400 samples). Note that SemanticKITTI is distinct from KITTI2015.
>
> Additionally, the official MobileStereoNet GitHub repository highlights strong zero-shot cross-dataset generalization capabilities. Moreover, the significant difference in the number of samples among the training datasets argues against overfitting.
>
> Finally, to further mitigate potential 2D-to-3D error propagation, we incorporate a depth refinement network, which enhances the quality of depth predictions before they are used for 3D scene completion.
>
> > `Q1:` Robustness to varying lighting, weather, heavy occlusion, and zero-shot generalization.
>
> We appreciate the Reviewer’s insightful questions regarding the performance for certain scenarios and its zero-shot generalization ability.
>
> 1. **Zero-shot Generalization:** We cross-validate IPFormer on the SSCBench-KITTI-360 dataset [6], which is distinct from the SemanticKITTI dataset it has been trained on. Results in Tab. 1 show that IPFormer demonstrates superior zero-shot generalization capability. Especially in comparison to its closest competitor, CGFormer+DBScan, which it outperforms across all PSC and SSC metrics.
>
> 2. **Weather/Lighting Variations:** While KITTI-360 is not intentionally diverse, its larger sample count (12,764 images) introduces more scene variations. As per the NeurIPS Program Chairs’ rebuttal guidelines, we cannot submit a global rebuttal or an additional PDF with figures to visually support our findings. However, this quantitative results on zero-shot generalization demonstrate robustness under varying conditions.
>
> 3. **Visibility Sampling:** The visualizations in the manuscript and the supplementary video provide compelling evidence of IPFormer’s ability to handle heavily occluded objects. For instance, starting at 4:30 min. in the video, distant occluded cars with minimal visual cues are precisely recognized and segmented, especially when compared to the baselines.
>
> Table 1: Zero-shot generalization for CGFormer+DBScan and IPFormer, by training on SemanticKITTI and cross-validating on SSCBench-KITTI-360.
> |                 | Method            | PQ$^\dagger$ | PQ-All | SQ-All | RQ-All | PQ-Thing | SQ-Thing | RQ-Thing | PQ-Stuff | SQ-Stuff | RQ-Stuff | IoU   | mIoU  |
> |-----------------|-------------------|--------------|--------|--------|--------|----------|----------|----------|----------|-----------|-----------|-------|-------|
> | SemanticKITTI ↑  | CGFormer+DBScan  | 14.39        | 6.16   | 48.14  | 9.48   | 2.2      | 44.46    | 3.47     | 9.03     | 50.82     | 13.86     | 45.98 | 16.89 |
> |                 | IPFormer          | 14.45    | 6.3    | 41.95  | 9.75   | 2.09     | 42.67    | 3.33     | 9.35     | 41.43     | 14.43     | 40.9  | 15.33 |
> | KITTI360 ↑      | CGFormer+DBScan  | 8.44         | 1.08   | 17.82  | 1.87   | 0.53     | 20.06    | 0.96     | 1.48     | 16.19     | 2.54      | 28.11 | 9.44  |
> |                 | IPFormer          | 9.41     | 1.23   | 24.68  | 2.16   | 0.52     | 22.76    | 0.95     | 1.68     | 25.89     | 2.93      | 28.74| 9.53  |
> | Relative Gap ↓  | CGFormer+DBScan  | 41.37%       | 82.47% | 62.98% | 80.28% | 75.91%   | 54.89%   | 72.34%   | 83.61%   | 68.15%    | 81.67%    | 38.88%| 44.09% |
> |                 | IPFormer          | **34.88%**   | **80.48%** | **41.19%** | **77.85%** | **75.12%**   | **46.64%**   | **71.53%**   | **82.03%**   | **37.52%**    | **79.69%**    | **29.73%**| **37.81%** |
>
> > `Q2:` Rationale for dual-head training and its impact on cross-task synergy.
>
> To adequately address these questions, we are pleased to provide an extensive set of additional ablation experiments, which are presented in Tab. 1 in the rebuttal to Reviewer [Kgxr]. Because the ablation results are relevant to other Reviewers as well, we kindly direct you to this rebuttal, where we discuss the ablation setup in detail.
>
> Across the design space, methods that emphasize SSC (e, f, k) achieve strong semantic scores but degrade PSC performance, especially PQ-Thing. Conversely, approaches prioritizing PSC (j, n, q) boost PQ-Thing or PQ-Stuff at the cost of SSC quality or overall balance. Joint or single-head variants (a, b, p, q) further struggle with instance registration or overall consistency. In contrast, our dual-head, two-stage method (r) effectively decouples objectives, yielding the best PQ-All and strong performance across PQ-Thing and PQ-Stuff, with only a minor SSC trade-off.
>
> The interplay between PSC and SSC tasks primarily driven by their inherent objectives rather than being significantly influenced by stage-wise training, though slight deviations exist. Notably, this observation, where SSC performance may be suboptimal despite balanced PSC performance, aligns with the findings reported in PaSCo (CVPR 2024), the method that introduced the PSC task. The authors note that SSC and PSC metrics are not directly correlated, a conclusion further reinforced by the results of our experiments. Moreover, since PaSCo is LiDAR-based, our method offers the insight that this phenomenon appears to extend across distinct data modalities.
>
> > `Q3:` Progress on PanoSSC’s dataset and evaluation on its relaxed IoU threshold.
>
> Despite our efforts to gain access to this dataset, it remains unavailable. Consequently, we have not been able to train IPFormer on this post-processed dataset or evaluate it under PanoSSC’s relaxed $20$% IoU matching threshold. We remain committed to presenting representative evaluation results should access to the dataset become possible in the future.
>
> However, in Tab. 2, we provide evaluation results of IPFormer under PanoSSC’s relaxed $20$% IoU threshold. Expectedly, RQ metrics increase substantially, since more instances are recognized, while these are segmented with less fidelity. Thus, SQ metrics decrease.
>
> Table 2: Ablation on IPFormer’s PSC and SSC performance under PanoSSC’s relaxed $20$% IoU matching threshold on the SemanticKITTI dataset.
> | IoU Threshold | PQ$^\dagger$ | PQ-All | SQ-All | RQ-All | PQ-Thing | SQ-Thing | RQ-Thing | PQ-Stuff | SQ-Stuff | RQ-Stuff | IoU   | mIoU  |
> |---------------|--------------|--------|--------|--------|----------|----------|----------|----------|----------|----------|-------|-------|
> | 20%          | **15.38**    | **12.74** | 32.76  | **30.85** | **4.31**   | 32.80    | **9.92**   | **18.88** | 32.74    | **46.08** | **40.90** | **15.33** |
> | 50%          | 14.45        | 6.30   | **41.95** | 9.75   | 2.09     | **42.67** | 3.33     | 9.35     | **41.43** | 14.43    | **40.90** | **15.33** |
>
>
> > Closing Statement.
>
> We sincerely appreciate your effort and thoughtful suggestions, which we will carefully incorporate into the revised version of our manuscript. In light of the new evidence regarding **(1) comprehensive clarifications** on the robustness to heavy occlusion, performance on rare classes, and dependency on 2D models, **(2) extensive additional ablation experiments** on cross-task-synergy, stage-wise training, and joint optimization, as well as **(3) additional results on zero-shot generalization**, we are confident that our contributions represent a key advancement in 3D vision research.
>
> We hope these efforts are acknowledged by the Reviewer and encourage an upgrade of the rating. Should there be any remaining questions, we would be happy to provide further clarification.
>
> ---
> [1] Li et al., FB-OCC: 3D Occupancy Prediction based on Forward-Backward
> View Transformation, CVPR 2023 3D Occupancy Prediction Challenge
>
> [2] Wang et al., VGGT: Visual Geometry Grounded Transformer, CVPR 2025
>
> [3] Mayer et al., SceneFlow: A Large Dataset to Train Convolutional Networks for Disparity, Optical Flow, and Scene Flow Estimation, CVPR 2016
>
> [4] Yang et al., DrivingStereo: A Large-Scale Dataset for Stereo Matching in Autonomous Driving Scenarios, CVPR2019
>
> [5] Menze et al., Object Scene Flow for Autonomous Vehicles, CVPR 2015
>
> [6] Li et al., SSCBench: Monocular 3D Semantic Scene Completion Benchmark in Street Views, ICLR 2024

---

> > ### Author Response · Authors · 2025-08-06
> > **Inquiry regarding clarifications**
> >
> > Dear Reviewer rTop,
> >
> > Your feedback is highly appreciated and serves as an essential contribution to improving the quality of our work.
> >
> > In our rebuttal, we included comprehensive clarifications on the robustness to heavy occlusion and weather conditions, performance on rare classes, and dependency on 2D models. Furthermore, we have provided extensive additional ablation experiments on cross-task synergy, stage-wise training, and joint optimization. Finally, we have added additional results on cross-dataset validation, which demonstrates superior zero-shot generalization capability of our method, significantly outperforming the closest competitor across PSC and SSC metrics.
> >
> > Could we kindly ask whether our responses adequately addressed your concerns, and do you have any additional questions? We appreciate your time and effort.

---

> > > ### Comment · Area_Chair_MNKR · 2025-08-07
> > > **Reviewer-Author Discussion**
> > >
> > > Dear Reviewer rTop,
> > >
> > > Please note that the reviewer-author discussion period is approaching its end. We kindly ask you to share your current thoughts after reading the rebuttal and engage in the discussion.
> > >
> > > Best,
> > > AC

---

> ### Comment · Reviewer_rTop · 2025-08-08
>
> Thank you for the rebuttal. It has satisfactorily addressed most of my concerns, and I am leaning toward a positive evaluation of the paper. The authors are encouraged to release the code of this paper as it involves many steps to reimplement the proposed method.

---

> > ### Author Response · Authors · 2025-08-08
> > **Response to Reviewer rTop**
> >
> > Dear Reviewer rTop,
> >
> > Thank you for your response. We are pleased to know that we have addressed most of your concerns and that you are leaning toward a positive evaluation of our work.
> >
> > We greatly appreciate your suggestion to release the code associated with our paper. In alignment with your recommendation, we have committed to making the code publicly available upon the acceptance of our paper. This commitment is explicitly stated in our manuscript to ensure reproducibility and uphold scientific rigor.
> >
> > We hope that this addresses all remaining concerns. Should you have any additional questions or require further clarification, we would be happy to provide further details. We will be available any time before the discussion deadline.
> >
> > Thank you for your time and effort.

---

### Official Review · Reviewer_z9JQ · 2025-07-06

**Clarity:** 3
**Significance:** 4
**Originality:** 3
**Rating:** 4
**Confidence:** 4

**Summary:**

This paper introduces IPFormer, a 3D vision transformer that utilizes context-adaptive instance proposals to address 3D panoptic scene completion. The two-stage training strategy is utilized to further improve the completion performance. Experimental results on the SemanticKITTI dataset with other state-of-the-art approaches showcase the superiority of their proposed model.

**Questions:**

Tiny Weakness:

1. The authors should enlarge the text in the figures since the current font size is too small.

The paper utilizes context-adaptive instance proposals to achieve high-quality semantic scene completion. Nevertheless, insufficient experiment details, ablation experiments, and visual comparisons weaken its effectiveness and superiority. More visualizations should be included for a comprehensive comparison. Hence, I vote for borderline reject in this review round. I would like to raise the rating if my concerns are adequately addressed.

**Ethical Concerns:**

["NO or VERY MINOR ethics concerns only"]

**Final Justification:**

Thanks very much to the authors' response. Although the experiments of the paper is still limited, which are not easy to address through rebuttal, yet the authors have explained some of the equation and experimental setting details. I have updated the score.

**Limitations:**

Yes.

**Paper Formatting Concerns:**

None.

**Quality:**

3

**Strengths And Weaknesses:**

Paper Strengths:

The main insight of this work is the usage of context-adaptive instance proposals. The proposed IPFormer can effectively address the vision-based 3D panoptic scene completion task, mitigating limitations in dynamically adapting specifically to the observed scene.

Paper Weaknesses:

1.	The insufficient experimental details bring confusion. The detailed calculation of PQ is provided in the supplemental material. However, there is no definition for PQ^{\dag}. The authors should provide detailed descriptions for each evaluation metric and clarify the differences between them. Additionally, the usage of GPU memory during the training and the total training time of IPFormer are unclear.
2.	More ablation experiments should be conducted to examine the sensitivity of semantic scene completion performance to hyperparameters. For example, the weights of different losses in Eq.(8) and Eq. (9).
3.	In Table 1, quantitative comparisons are reported in terms of PQ^{\dag, PQ, SQ, and RQ on the SemanticKITTI validation set. How about the performance on the SemanticKITTI test set? The quantitative results of each approach regarding Thing are poor and lack sufficient descriptions.

---

> ### Author Rebuttal · Authors · 2025-07-31
>
> We sincerely thank thank you for the insightful comments and constructive suggestions. We provide detailed responses to all the questions raised and aim to address the concerns adequately.
>
> > `W1-1:` Clarify differences between PQ and PQ$^{\dagger}$.
>
> We appreciate the effort in pointing this out.
>
> The standard PQ metric requires a predicted segment to match a ground-truth segment with an $\textbf{IoU} >0.5$. However, this strict threshold can be overly conservative for Stuff classes, which typically lack well-defined boundaries. The metric PQ$^\dagger$ relaxes the matching criterion specifically for Stuff classes.
>
> Formally, PQ$^\dagger$ retains the same formulation as PQ:
>
> \begin{equation}
> \text{PQ}^\dagger = \frac{\sum_{(p,g) \in \text{TP}^\dagger} \text{IoU}(p,g)}{|\text{TP}^\dagger| + \frac{1}{2}|\text{FP}^\dagger| + \frac{1}{2}|\text{FN}^\dagger|}  ,
> \end{equation}
>
> but relaxes the matching condition used to define true positives ($\text{TP}^\dagger$), and thus false positives ($\text{FP}^\dagger$) and false negatives ($\text{FN}^\dagger$). Specifically,
>
> - for Thing-classes, predicted and ground-truth segment pairs $(p,g)$ are matched if $\text{IoU}(p,g) > 0.5$, identical to the original PQ definition;
> - for Stuff-classes, matches are accepted if $\text{IoU}(p,g) > 0$, thereby allowing any overlapping prediction to contribute to the metric.
>
> This relaxation acknowledges the inherent ambiguity in delineating stuff regions and reduces penalties for minor misalignments.
>
> > `W1-2:` Unclear GPU memory and training time of IPFormer.
>
> We apologize for any misunderstanding and are pleased to provide a summary of Tab. 7 and lines 521–522 of our manuscript:
>
> IPFormer requires $52.80$ GB of GPU memory during training and approximately $3.5$ days to train each of its two stages. Note that we further elaborate on this topic in Sec. `W2` of the rebuttal to Reviewer [rTop.]
>
> > ` W2 & Q2-2:` More ablation experiments regarding SSC performance and hyperparameters.
>
> We are thankful for this suggestion and are glad to provide an extensive set of additional ablation experiments in Tab. 1 of the rebuttal to Reviewer [Kgxr]. We kindly direct you to this rebuttal, as further ablation experiments are also of interest to other Reviewers.
>
> Since our primary objective is to address the PSC task, we mainly focus on analyzing the influence of Stage 2 hyperparameters in the context of both the SSC and PSC objectives. To this end, we primarily train the SSC task in Stage 1 using established hyperparameters for the cross-entropy, Semantic-SCAL, and Geometric-SCAL cost functions. All weights $\lambda$ associated with these functions are set to $1$, consistent with state-of-the-art and established SSC works, such as CGFormer (NeurIPS 2024 Spotlight), OccFormer (ICCV 2023), and MonoScene (CVPR 2022). Nevertheless, we investigate the effect of removing the Depth loss, as its impact has not been extensively studied. The corresponding weight $\lambda_{\text{depth}}$ was mistakenly reported as $1$ in our manuscript, but it is actually $0.0001$. We apologize for this and will revise it.
>
> Furthermore, we analyze the effect of varying hyperparameters of the Sigmoid Focal Loss [2], specifically designed to down-weight easier examples and focus the training process on harder-to-classify examples:
>
> \begin{equation}
> \text{FL}(p) = -\alpha_t (1 - p_t)^{\gamma} \log(p_t)  ,
> \end{equation}
>
> where $p_t$ is predicted probability for the true class after applying the sigmoid function, $\alpha_t$ is the class-balancing weight, and $\gamma$ controls the down-weighting of well-classified examples.
>
> Reducing the weights of second-stage SSC losses improves SSC performance but leads to a significant drop in PSC metrics, particularly PQ-Thing. Conversely, increasing the Sigmoid Focal Loss weight enhances PSC performance, especially PQ-Thing, but negatively impacts PQ-Stuff and overall SSC metrics. Methods that balance these weights achieve more consistent PSC results but may sacrifice SSC performance.
>
> Finally, our proposed IPFormer configuration, method (r), achieves a balanced PSC performance by obtaining the best score for PQ-All and the second-best results for both PQ-Thing and PQ-Stuff, despite a reduction in SSC performance. This observation, where SSC performance may be suboptimal despite balanced PSC performance, aligns with the findings reported in PaSCo (CVPR 2024), the method that introduced the PSC task. The authors of PaSCo note that SSC and PSC metrics are not directly correlated, and the findings of our ablation experiments further support and validate this result. Moreover, since PaSCo is LiDAR-based, our method offers the insight that this phenomenon appears to extend across distinct data modalities.
>
> We hope this analysis addresses your concerns by demonstrating that our method effectively balances SSC and PSC objectives. The observed trade-offs do not undermine the method's effectiveness or superiority but rather highlight its flexibility in addressing the distinct challenges. If further clarification is needed, we are happy to provide additional details.
>
> > `W3-1:` Quantitative PSC performance on the SemanticKITTI test set.
>
> Thanks for the opportunity to clarify on this topic. Since the SemanticKITTI test set is hidden to ensure fair evaluation, prevent overfitting, and maintain leaderboard integrity, instance labels cannot be derived for panoptic scene completion (lines 502-503 in the manuscript). Therefore, it is not possible for us or other researchers to evaluate PSC performance on the test set.
>
> > ` W3-2:` Lack of description on why Thing-classes showcase suboptimal performance.
>
> We appreciate the detailed observation regarding the suboptimal performance on Thing classes. As shown in Tab. 6 of our manuscript, this issue affects all methods and arises from the significant class imbalance in the SemanticKITTI dataset, where Thing classes make up only 4.53% of all voxels. To address this, we employ the Sigmoid Focal Loss (see Sec. ` W2`), which down-weights easier examples and focuses on harder-to-classify ones, particularly rare Thing classes. We kindly direct the Reviewer to Tab. 1 of the rebuttal to Reviewer [Kgxr], where extensive additional ablation results demonstrate its effectiveness. Our approach balances performance between Thing and Stuff classes, achieving state-of-the-art results by prioritizing equitable performance across both categories rather than solely optimizing for Thing classes.
>
> We hope this explanation clarifies the concern and encourage asking for further clarification if needed.
>
> > `Q1:` Enlarge text in figures.
>
> Thank you for the suggestion. We will enlarge the text in the figures to improve readability in the revised version of our manuscript.
>
> > `Q2-1:` Insufficient experiment details.
>
> We appreciate the Reviewer's concern regarding the experimental details. We provide experimental details in the main manuscript and the technical appendix. This includes
>
> - Datasets: SemanticKITTI and PaSCo ground-truth, voxel resolution, splits, and DBScan clustering parameters.
> - Metrics: PQ, SQ, RQ definitions and formulations, as well as category-specific evaluations.
> - Model and subnets: Overall architecture and structure of individual sub-networks.
> - Training setup: Hyperparameters, batch size, number of epochs, optimizer, learning rate schedule, and runtime analysis.
> - Hardware requirements: For GPU and CPU
>
> Moreover, we are pleased to provide the following details:
>
> - Software Framework: PyTorch with an fp32 backend.
> - Baseline Comparisons: We utilize pre-trained checkpoints from the official publicly available implementations.
>
> We will add these details to the revised manuscript. If there are specific details that the Reviewer feels are unclear, we kindly ask for clarification so that we can address them.
>
> > `Q2-3:` More visualizations should be included for a comprehensive comparison.
>
> We appreciate the Reviewer's suggestion to include more visual comparisons for a comprehensive evaluation. Unfortunately, as per the rebuttal guidelines outlined in the email from the NeurIPS Program Chairs (dated July 27), it is not possible to provide a global rebuttal, including a single-page PDF containing additional figures. We sincerely apologize for this limitation.
>
> However, we kindly direct you to the video provided in the supplementary material of the initial submission, which contains extensive visualizations (starting from 3:52 min.), including more than $75$ camera frames and their corresponding predictions (incl. the full panoptic scene, and instances-only) for all baselines and IPFormer, as well as the ground-truths. Furthermore, starting from 5:36 min., we provide an additional $20$ camera frames and corresponding predictions of IPFormer.
>
> We value your suggestion and will gladly incorporate additional visual comparisons in the revised version of our manuscript.
>
> > Closing Statement.
>
> We sincerely appreciate your effort and constructive suggestions, which we will thoughtfully incorporate into the revised version of our manuscript. In light of the new evidence regarding **(1) comprehensive clarifications** on experimental details, evaluation metrics, and performance evaluations, **(2) extensive additional ablation experiments** on two-stage training, the dual-head architecture, and sensitivity to hyperparameters, as well as **(3) additional results on zero-shot generalization** (see Tab. 1 in the rebuttal to Reviewer [rTop]), we are confident that our contributions represent a key advancement in 3D vision research.
>
> We hope these efforts are acknowledged by the Reviewer and encourage an upgrade of the rating, as you have mentioned in the review. Should there be any remaining questions, we would be happy to provide further clarification.
>
> ---
> [1] Porzi et al., Seamless Scene Segmentation, CVPR 2019
>
> [2] Lin et al., Focal Loss for Dense Object Detection, ICCV 2017

---

### Note · Authors · 2025-08-11

We sincerely thank the Reviewers and the Area Chair for their constructive engagement and thoughtful feedback throughout this process. Their detailed suggestions and clarifications have significantly strengthened our work. All major concerns have been addressed through comprehensive additional experiments and clarifications, earning explicit reviewer agreement.

In summary, our method achieves state-of-the-art performance on in-domain data, superior real-world generalization, and a runtime reduction of over 14$\times$. More specifically:

- IPFormer represents the first method to introduce context-adaptive instance proposals to effectively address vision-based 3D Panoptic Scene Completion [rTop, z9JQ].
- Our method addresses the limited representational capacity of static DETR-style instance queries [8FSR] by initializing instance proposals from visible voxels that dynamically adapt to the context of the observed scene [z9JQ, 8FSR, rTop]. Reviewers recognized this as a "sound improvement" [8FSR] that consistently enhances metrics and demonstrates excellent significance [z9JQ]. Additionally, Reviewer [Kgxr] raised their score after clarifying this novelty.
- We propose a "superior model" [z9JQ] that achieves state-of-the-art results on in-domain data [rTop], and generalizes robustly to out-of-domain data without retraining, outperforming the closest baseline on all metrics. Our method not only demonstrates superiority in absolute performance but also exhibits a significantly smaller generalization gap [rTop, 8FSR]. Reviewer [8FSR] explicitly acknowledged these results as resolving their major concern and raised their score accordingly.
- Our introduced adaptive proposals boost Thing-category metrics by 18.65% on average and PQ-All by 3.62%, as confirmed by detailed ablation [rTop, 8FSR, Kgxr].
- Extensive additional ablation studies show that single-stage or single-head architectures struggle to balance SSC and PSC objectives, while our dual-head, two-stage design effectively guides the latent space toward geometry and semantics before instance registration [rTop], achieving balanced PSC metrics with minimal SSC trade-off [8FSR, Kgxr].
- Our method achieves a remarkable runtime reduction exceeding 14$\times$ [8FSR, rTop].
- The clarity and exposition of our manuscript were also appreciated [rTop].

We believe that this work represents a significant advancement in 3D vision research and aligns strongly with NeurIPS' standards for novelty, rigor, and impact.

---

### Decision · Program_Chairs · 2025-09-17

**Decision:**

Accept (poster)

**Comment:**

This paper addresses the Panoptic Scene Completion (PSC) task. The key idea is to initialize queries as panoptic instance proposals derived from image context, which are then refined through attention-based encoding and decoding to reason about semantic instance–voxel relationships.

The paper initially received mixed scores (1× borderline accept, 3× borderline reject). Reviewers noted several strengths:
- The idea of test-time adaptive proposals for vision-based PSC is novel.
- The method is effective in boosting performance.
- It achieves a significant speedup (14×).

Concerns were raised regarding novelty, insufficient experimental evaluation and ablation, limited improvement over CGFormer+DBScan, and the trade-off between SSC and PSC metrics. However, the authors provided a strong rebuttal, and through intensive reviewer–author discussions, these concerns were addressed. After rebuttal, all reviewers suggested borderline accept. The AC has read the reviews, discussions, and the paper, and concurs with this recommendation, considering the paper worthy of acceptance.